# Mitigating Group Bias in Federated Learning: Beyond Local Fairness

**Ganghua Wang**                                             *wang9019@umn.edu*
*School of Statistics*
*University of Minnesota*

**Ali Payani**                                               *apayani@cisco.com*
*Cisco Research*

**Myungjin Lee**                                             *myungjle@cisco.com*
*Cisco Research*

**Ramana Kompella**                                          *rkompell@cisco.com*
*Cisco Research*

**Reviewed on OpenReview:** *https://openreview.net/forum?id=ANXoddnzct*

## Abstract

The issue of group fairness in machine learning models, where certain sub-populations or groups are favored over others, has been recognized for some time. While many mitigation strategies have been proposed in centralized learning, many of these methods are not directly applicable in federated learning, where data is privately stored on multiple clients. To address this, many proposals try to mitigate bias at the level of clients before aggregation, which we call locally fair training. However, the effectiveness of these approaches is not well understood. In this work, we investigate the theoretical foundation of locally fair training by studying the relationship between global model fairness and local model fairness. Additionally, we prove that for a broad class of fairness metrics, the global model's fairness can be obtained using only summary statistics from local clients. Based on that, we propose a globally fair training algorithm that optimizes the fairness-regularized empirical loss. Real-data experiments demonstrate the promising performance of our proposed approach for enhancing fairness while retaining high accuracy compared to locally fair training methods.

## 1 Introduction

As edge devices such as mobile phones and wearable devices have been heavily involved in our daily life, leveraging the enormous data collected by those devices and their computational resources to train machine learning models has attracted increasing research interest. One challenge is that datasets collected by different devices are often forbidden to be shared due to communication costs and privacy concerns. Thus, classical centralized learning, where data is gathered and stored in a central database, is not suitable. To address those challenges, federated learning (McMahan et al., 2017; Konečný et al., 2016) has been proposed to train models in a decentralized manner. In federated learning, a global model is distributed to multiple clients, or edge devices, which update the model using their own data and send the updated model back to a central server. The server then aggregates the updated models to obtain a new global model and the process is repeated.

While significant progress has been made in the theory and application of federated learning (Li et al., 2020b), most research has focused on improving the prediction accuracy of the global model. As these models are increasingly being used in areas that have a direct impact on people's lives, such as healthcare, finance, and criminal justice (Berk, 2019; Barocas & Selbst, 2016; Becker, 2010), the ethical implications of these models

have attracted a lot of attention. In particular, it is crucial for the learned model to treat different groups in the population equitably. Nevertheless, it has been recognized that without careful consideration of group fairness [1] the learned model may be biased (Guion, 1966; Cleary, 1968; Hutchinson & Mitchell, 2019) . For example, the COMPAS algorithm (Angwin et al., 2016), which assigns recidivism risk scores to defendants based on their criminal history and demographic attributes, was found to have a significantly higher false positive rate for black defendants than white defendants, thereby violating the principle of equity on the basis of race. This highlights the risk of similar issues to arise in other applications such as university admissions and job screenings, which can negatively impact diversity and ultimately harm the society.

Though bias mitigation has been extensively studied in the centralized setting (Kamiran & Calders, 2012; Kamishima et al., 2012; Zafar et al., 2017; Zhang et al., 2018; Hardt et al., 2016; Lohia et al., 2019; Caton & Haas, 2020), it remains under-explored for federated learning. Many of the currently proposed algorithms try to reduce the global model's bias by minimizing the bias of local models (Abay et al., 2020; Ezzeldin et al., 2021), hereinafter referred to as *locally fair training* (LFT). Because the global model is the average of local models, LFT hopes the global model is fair as long as local models are fair. However, theoretical understanding of LFT is limited, such as under what conditions LFT is effective. One main challenge of analysis is obtaining the fairness measure for the global model without sharing original data across devices. In this work, we tackle this challenge for a particular class of fairness metrics.

## 1.1 Our Contributions

Our contributions are three-fold as summarized below.

1. We formulate the definitions of group-based and proper group-based fairness metrics. For proper group-based metrics, the global fairness value can be expressed as a function of fairness-related statistics calculated by local clients solely. This property enables us to calculate the global fairness value without directly accessing local datasets. In particular, those fairness-related statistics are not local fairness values, distinct from all existing works.

2. Under our proposed group-based fairness notion, we investigate the relationship between the fairness of local and global models. We find that global fairness and local fairness do not imply each other in general. Nevertheless, for proper group-based fairness defined in Section 4, the global fairness value is controlled by the local fairness values and the data heterogeneity level. This result explains the success of LFT methods in the setting of near-homogeneous clients for common fairness metrics, such as demographic parity and equal opportunity. It supports the results in Chu et al. (2021) that local fairness implies global fairness for homogeneous clients. Our work also provides a fairness analysis based on data heterogeneity, complementing the information theory driven analysis presented in Hamman & Dutta (2023).

3. We propose a globally fair training method named FedGFT for proper group-based metrics. FedGFT goes beyond LFT by directly solving a regularized objective function consisting of the empirical prediction loss and a penalty term for fairness. Additionally, it applies to clients with arbitrary data heterogeneity. Numerical experiments on multiple datasets show that FedGFT significantly reduces the bias of the global model while retaining high prediction accuracy.

## 1.2 Related Work

**Fair federated learning methods.** For bias mitigation methods in federated learning, one popular approach is locally fair training (LFT). For example, Abay et al. (2020)('**LRW**'), Chu et al. (2021), and Ezzeldin et al. (2021) ('**FairFed**') propose to train local models by applying centralized bias mitigation methods, such as reweighing the dataset to balance the group distribution (Kamiran & Calders, 2012), and adding a constraint or a penalizing term of fairness on the optimization objective function (Zafar et al., 2017; Kamishima et al., 2012).

---

[1]In this paper, improving model fairness is also understood as decreasing model bias.

Works on handling fairness in federated learning other than LFT have also been emerging. Zhang et al. (2018) used a reinforcement learning approach to select clients that participate in the training with the highest local fairness and accuracy. Du et al. (2021) proposed to add a global fairness constraint to the agnostic federated learning formulation. Rodríguez-Gálvez et al. (2021) proposed to solve a fairness-constrained optimization problem. Zeng et al. (2021) ('**FedFB**') proposed to solve a bi-level optimization problem with the outer loop adaptively choosing fair batch representation of the training data. Mehrabi et al. (2022) assumed a validation dataset is available for evaluating local fairness and assigned higher weights to fairer clients. Hu et al. (2024) proposed an algorithm with proved fairness guarantee under the bounded group loss, a privacy notion that can be viewed as a relaxation of demographic parity or equalized odds.

**Theoretical understanding of fair federated learning methods.** There have been a few works to understand locally fair training recently. Zeng et al. (2021) showed that training locally fair models with federated learning is better than assembling locally fair models without iterative server-client updates, but worse than centralized training. Chu et al. (2021) proved that for homogeneous clients and a specific fairness metric, locally fair training yields a global model with a fairness guarantee.

Recently, Hamman & Dutta (2023) explore the connections between global and local fairness in FL for statistical disparity from the perspective of information theory. Specifically, motivated from the definition of statistical disparity, Hamman & Dutta (2023) define global disparity as the mutual information between the model response and the true label, and defines local disparity as the conditional mutual information between the model response and the true label given the sensitive attribute. Benefited from the specific form of those two definitions, Hamman & Dutta (2023) derive the sufficient and necessary condition that local disparity implies global disparity, assuming the underlying data distribution is known. In contrast, our work focuses on a broad class of fairness metrics (see Definition 1), including statistical disparity, equal opportunities, calibration, and more. We show that global fairness in general does not imply local fairness, and vice versa. Moreover, when fairness metric is proper, we provide an upper bound of global fairness by local fairness and data heterogeneity.

**Comparison with existing works.** Our main distinction from existing methods lies in our theory-inspired approach to minimizing global fairness, namely solving a global objective function that combines empirical loss and a regularization term for global fairness. Compared to FedGFT, LFT seeks a weighted sum of locally fair models, thus its minimization objective does not exactly match the global fairness objective. As a result, LFT lacks theoretical support for achieving true global fairness, potentially leading to suboptimal outcomes. On the other hand, established approaches to directly optimize global fairness often involve solving constrained optimization problems, necessitating the implementation of bi-level optimization algorithms (e.g. Rodríguez-Gálvez et al., 2021; Zeng et al., 2021; Chu et al., 2021; Du et al., 2021). In contrast, FedGFT streamlines the optimization process due to the form of our penalized objective and insights from Theorem 2. Therefore, FedGFT offers advantages such as: (1) No Bi-level Optimization. By dispensing with the need for bi-level optimization, our algorithm seamlessly integrates with conventional federated learning algorithms by slightly adapting the local objective. (2) Minimal Communication Overhead. Our proposed algorithm incurs minimal communication costs related to fairness metrics. FedGFT only requires real numbers as additional communication cost, instead of requiring extra parameter gradients such as (Chu et al., 2021; Rodríguez-Gálvez et al., 2021).

## 2 Preliminaries

### 2.1 Federated Learning

There is a large body of literature on federated learning (Li et al., 2020a) since proposed by (Konečný et al., 2016; McMahan et al., 2017). It aims to train a global machine learning model while keeping the training data privately on edge devices, also named local clients. Suppose there are $K$ clients in total, and the $k$-th client owns $n_k$ training data $\{\boldsymbol{X}_k^{(i)}, Y_k^{(i)}\}_{i=1}^{n_k}$, where $\boldsymbol{X}$ is the predictor and $Y$ is the response. Let $l(\cdot, \cdot)$ be a

loss function, federated learning aims to solve the following empirical risk minimization problem:

$$\min_{\theta} \sum_{k=1}^{K} \frac{n_k}{n} L_k(\theta), \text{ where } L_k(\theta) = \frac{1}{n_k} \sum_{i=1}^{n_k} l(f(\boldsymbol{X}_k^{(i)}; \theta), Y_k^{(i)}), n = \sum_{k=1}^{K} n_k. \tag{1}$$

Here, $L_k(\theta)$ is the empirical risk of the $k$-th client, $f(\cdot; \theta)$ is a parameterized model.

The original idea of federated learning is training the model on each client using its local dataset for several updating steps, then aggregating the local models on the central server to obtain a global model, and repeating the above procedure until meeting the terminating conditions. More specifically, at each communication round $t$, the server first propagates the parameters $\theta^t$ of the current global model to the clients. Then, each client will perform $E$ epochs of local updates to get $\theta_k^{t,E}, k = 1, \ldots, K$. Finally, the server will aggregate $\theta_k^{t,E}$'s to a new global model with parameter $\theta^{t+1}$.

### 2.2 Group Fairness

There are many different interpretations of fairness (Caton & Haas, 2020; Mehrabi et al., 2021; Li et al., 2021; Yue et al., 2023; Papadaki et al., 2022). In this paper, we focus on group fairness, which ensures that the model will not have discriminatory behavior towards certain groups. For simplicity, we consider a binary classification task with the outcome $Y \in \{0, 1\}$, and sensitive group $A \in \{0, 1\}$. There are two major categories of group fairness quantification (Corbett-Davies & Goel, 2018). The first category is based on the classification parity, which means a measure of the prediction error is equal across different groups. For example, statistical parity (Kamishima et al., 2012; Feldman et al., 2015), also known as demographic parity, requires that the distribution of the prediction $\widehat{Y}$ conditional on the sensitive group is the same. In other words, $\mathbb{P}(\widehat{Y} = 1 | A = 0) = \mathbb{P}(\widehat{Y} = 1 | A = 1)$. Another example is equal opportunity (Hardt et al., 2016), which requires the same true positive rate across groups, i.e., $\mathbb{P}(\widehat{Y} = 1 | A = 0, Y = 1) = \mathbb{P}(\widehat{Y} = 1 | A = 1, Y = 1)$. The second category is calibration (Pleiss et al., 2017). A model is well-calibrated or achieves test fairness if the true outcome is independent of the group given the predicted value. We note that different fairness definitions may be incompatible; actually, it is impossible to achieve multiple fairness goals simultaneously (Kleinberg et al., 2016).

## 3 Problem Formulation

This paper considers a binary classification task with outcome $Y \in \{0, 1\}$. Suppose the predictors $\boldsymbol{X} = (X_1, \ldots, X_p)^{\mathrm{T}} \in \mathbb{R}^p$ are $p$-dimensional variables. Without loss of generality, we assume the first predictor to be the sensitive attribute as $A = X_1 \in \{0, 1\}$, and other predictors are non-sensitive. We consider a heterogeneous scenario that there are $K$ clients ($K > 1$), and the $k$-th client's training data $\{\boldsymbol{X}_k^{(i)}, Y_k^{(i)}\}_{i=1}^{n_k}$ is IID generated from a distribution $\mathcal{D}_k$. The empirical distribution of $\{\boldsymbol{X}_k^{(i)}, Y_k^{(i)}\}_{i=1}^{n_k}$ is denoted as $\widehat{\mathcal{D}}_k$. Our goal is to learn a function $f : \mathbb{R}^p \to [0, 1]$ from data, where $f(\boldsymbol{X})$ is regarded as the predicted probability of $\mathbb{P}(Y = 1 | X)$. The accuracy of the learned function $f$ is evaluated by the prediction risk $\mathbb{E}\{l(f(\boldsymbol{X}), Y)\}$, where $\mathbb{E}$ denotes expectation, and $l(\cdot, \cdot)$ is a loss function, such as the cross entropy loss. As for the fairness measure, we define the following group-based fairness metrics.

**Definition 1** (Group-based fairness metrics)**.** $F(f, \mathcal{D})$ is a group-based fairness metric if it is in the form of

$$F(f, \mathcal{D}) = \left| \frac{a(f, \mathcal{D})}{b(f, \mathcal{D})} - \frac{c(f, \mathcal{D})}{d(f, \mathcal{D})} \right|,$$

where $a(f, \mathcal{D})$ and $b(f, \mathcal{D})$ are some expectations on the event $\{A = 0\}$, $c(f, \mathcal{D}), d(f, \mathcal{D})$ are some expectations on the event $\{A = 1\}$. Moreover, we have the range of $a, b, c, d, a/b$ and $c/d$ be $[0, 1]$, where $a, b, c, d$ stands for four functions omitting the arguments.

The concept of group-based fairness metrics are motivated by the observation that many widely used group fairness metrics adopt the form of the disparity of model performance among different groups, such as the confusion-matrix based probabilities (Caton & Haas, 2020; Kim et al., 2020). Taking statistical disparity

Table 1: The associated functions of three fairness metrics.

| Metrics | $a(f, \mathcal{D})$ | $b(f, \mathcal{D})$ | $c(f, \mathcal{D})$ | $d(f, \mathcal{D})$ |
|---|---|---|---|---|
| Statistical Parity | $\mathbb{P}(\widehat{Y} = 1, A = 0)$ | $\mathbb{P}(A = 0)$ | $\mathbb{P}(\widehat{Y} = 1, A = 1)$ | $\mathbb{P}(A = 1)$ |
| Equal opportunity | $\mathbb{P}(\widehat{Y} = 1, Y = 1, A = 0)$ | $\mathbb{P}(Y = 1, A = 0)$ | $\mathbb{P}(\widehat{Y} = 1, Y = 1, A = 1)$ | $\mathbb{P}(Y = 1, A = 1)$ |
| Well-Calibration | $\mathbb{P}(Y = 1, \widehat{Y} = 1, A = 0)$ | $\mathbb{P}(\widehat{Y} = 1, A = 0)$ | $\mathbb{P}(Y = 1, \widehat{Y} = 1, A = 1)$ | $\mathbb{P}(\widehat{Y} = 1, A = 1)$ |

for example, it is the difference of the probability being predicted as positive for individuals from different groups. In other words, when $A$ is a binary group variable taking values in 0 and 1, the statistical disparity is measured by $|P(\widehat{Y} = 1|A = 0) - P(\widehat{Y} = 1|A = 1)|$, where $\widehat{Y}$ is the prediction by model $f$. We thus propose that the model performance given a specific group $s$ can be typically expressed as a conditional expectation on the event that $A = s$, namely $E(g(\widehat{Y})|A = s)$ for any evaluation function $g$. In the case of statistical disparity, $g(\widehat{Y}) = 1_{\widehat{Y}=1}$ and $1_{(\cdot)}$ is the indicator function.

Then, why our Definition 1 takes form of the difference of two ratios? There are two critical reasons. First, the Bayes theorem shows that $E(g(\widehat{Y})|A = s)$ can always be written as the ratio of two expectations, therefore a group fairness metric can be naturally written as the form in Definition 1. Second, the increase in the degree of freedom (the additionally introduced $b$ and $d$ in Definition 1) enables a finer grid analysis on the relationship between global and local fairness. In particular, we will show in the following sections that while global fairness and local fairness are not aligned for general group-fairness metric, they are closely related if $b$ and $d$ are irrelevant to the model $f$.

In short, it is the first time a theoretical formulation is given to group-based fairness metrics. Clearly, a smaller $F(f, \mathcal{D})$ indicates higher model fairness and smaller model bias. Definition 1 includes many common measures, such as the following three. We can verify this by checking Table 1, with full details in supplementary document.

**Statistical Parity (SP).** It is defined as $F(f, \mathcal{D}) = |\mathbb{P}(\widehat{Y} = 1|A = 0) - \mathbb{P}(\widehat{Y} = 1|A = 1)|$.

**Equal Opportunity (EOP).** $F(f, \mathcal{D}) = |\mathbb{P}(\widehat{Y} = 1|A = 0, Y = 1) - \mathbb{P}(\widehat{Y} = 1|A = 1, Y = 1)|$.

**Well-Calibration.** $F(f, \mathcal{D}) = |\mathbb{P}(Y = 1|A = 0, \widehat{Y} = 1) - \mathbb{P}(Y = 1|A = 1, \widehat{Y} = 1)|$.

In practice, since the true underlying distribution is typically unknown, we take the empirical estimation $F(f, \widehat{\mathcal{D}}_k)$ as a surrogate for the local fairness of the $k$-th client, and use $F(f, \widehat{\mathcal{D}})$ as the global fairness, where $\widehat{\mathcal{D}} = \sum_{i=1}^{K} w_k \widehat{\mathcal{D}}_k, w_k = n_k/n, \ n = \sum_{i=1}^{k} n_k$.

Recall that the learned function $f$ is expected to be both accurate (with respect to the classification task) and fair (with respect to the sensitive group $A$). Locally fair training is one approach to extend bias mitigation methods from the centralized setting to the federated learning setting. Essentially, it minimizes the bias of each local client at each communication round and expects that the aggregation of locally fair models will yield a globally fair model. To better understand the effectiveness of locally fair training methods, we are going to study the following two fundamental questions in next sections:

1. What is the relationship between fairness of local models and the global model?

2. Is there an algorithm that directly targets improving global fairness?

The answer to the first question is local fairness does not imply global fairness in general. Nevertheless, for a proper group-based fairness metric, which is defined in Section 4, we show that global fairness can be controlled by local fairness and data heterogeneity. To our best knowledge, this is the first work to systematically study the relationship between local and global fairness. As for the second question, we propose such an algorithm called FedGFT in Section 5.

## 4 Locally Fair Training

This section explores the relationship between local and global fairness, which helps us in analyzing the locally fair training methods. The idea of minimizing the biases of local models is appealing at the first glance, based on an intuition that global fairness will be guaranteed if all local models are fair. Chu et al. (2021) proved that this intuition is true for homogeneous clients and a special fairness metric named accuracy disparity. However, we show that it does not hold in general. All proofs are included in Appendix.

**Theorem 2** (In general, Global $\neq$ local). *Suppose $F$ is a group-based fairness metric. For any $0 \leq C \leq 1$, there exist a model $f$ and local data distributions $\{\widehat{\mathcal{D}}_k, k = 1, \ldots, K\}$ such that $F(f, \widehat{\mathcal{D}}_k) = 0$ for all $k$, and $F(f, \widehat{\mathcal{D}}) \geq C$. Conversely, for any $0 \leq C \leq 1$, there also exists another set of $f$ and $\{\widehat{\mathcal{D}}_k, k = 1, \ldots, K\}$ such that $F(f, \widehat{\mathcal{D}}) = 0$, and $F(f, \widehat{\mathcal{D}}_k) \geq C$ for all $k$.*

**Corollary 3.** *Suppose $F$ is a group-based fairness metric, then $F(f, \widehat{\mathcal{D}})$ cannot be written as a linear combination of $\{F(f, \widehat{\mathcal{D}}_k), k = 1, \ldots, K\}$.*

Theorem 2 means that locally fair models do not imply a fair global model and vice versa. Therefore, locally fair training methods are not always effective in general. Additionally, Corollary 3 indicates that global fairness cannot be obtained from a simple average of local fairness. Simpson's paradox (Blyth, 1972; Bickel et al., 1975) is an excellent example to illustrate that the local property cannot represent that of the global, as shown in Table 2. Suppose a college has two departments, A and B, which accept applications from high school students. Here, gender is the sensitive group, and each department is considered a client. Although the acceptance rate is the same between males and females (i.e., SP is zero) for both departments, the overall acceptance rate is significantly biased toward males.

Table 2: The admission example of gender bias. 'APPL' and 'ACPT' stands for applicants and acceptance rate.

| Dept. | Female | | Male | |
|---|---|---|---|---|
| | APPL | ACPT | APPL | ACPT |
| A | 90 | 20% | 10 | 20% |
| B | 10 | 80% | 90 | 80% |
| Total | 100 | 26% | 100 | 74% |

The major factor that may lead to the failure of LFT is data heterogeneity, as revealed in the following two theorems.

**Theorem 4.** *Suppose $F$ is a group-based fairness metric, $f$ is non-degenerated, and $\mathcal{D} = \sum_{i=1}^{K} w_k \mathcal{D}_k$, where $w_k$ is the aggregation weight. A necessary and sufficient condition for $F(f, \mathcal{D}) = 0$ holds for any $w_k$ is that there exists a constant $C$ such that $a_k/b_k = c_k/d_k = C$ for all $k$, where $g_k = g(f, \mathcal{D}_k)$ for $g \in \{a, b, c, d\}$.*

Theorem 4 implies that if the global model is perfectly unbiased regardless of sample sizes of clients, then all local models are also unbiased. Note that this is assured if the data distributions of different clients are homogeneous. Inspired by Theorem 4, a natural idea to evaluate data heterogeneity is the maximum difference of $a_k/b_k$ (also $c_k/d_k$) among all clients. However, those quantities involve the global model $f$, which is unknown before the training. Thus, we introduce the following concepts to decouple with $f$.

**Definition 5** (Proper Group-based fairness metrics). *A group-based fairness metric $F(f, \mathcal{D})$ is proper if the corresponding $b(f, \mathcal{D})$ and $d(f, \mathcal{D})$ are degenerated with respect to $f$. In other words, there exist a function $b'$ such that $b(f, \mathcal{D}) = b'(\mathcal{D})$, and similarly for $d(f, \mathcal{D})$.*

**Definition 6** (Data heterogeneity with respect to $F$). *For a proper group-based fairness metric $F$, let $b = \sum_k w_k b_k$ and $d = \sum_k w_k d_k$. The data heterogeneity coefficient is defined as*

$$\mathrm{DH}(\{\widehat{\mathcal{D}}_k, k = 1, \ldots, K\}) = \max_k \left| \frac{d}{b} \frac{b_k}{d_k} - 1 \right|.$$

Many fairness measures are proper such as SP and EOP, while calibration is not proper, as indicated by Table 1. For proper metrics, DH measures the relative variation of two data-determined statistics $b_k$ and $d_k$, hence reflects the influence of data heterogeneity. More importantly, DH relates the global and local fairness as follows.

**Theorem 7** (Near IID, local implies global). *Suppose $F$ is proper, the data heterogeneity coefficient of clients' data is $\beta$, and $F(f, \widehat{\mathcal{D}}_k) \leq \alpha$ for all $k$, then $F(f, \widehat{\mathcal{D}}) \leq \alpha + \beta$.*

Theorem 7 shows that the global fairness is upper-bounded by the local fairness and data heterogeneity level for proper group-based fairness metrics. This upper bound is tight when data heterogeneity level $\beta$ is small. On the one hand, it justifies the success of locally fair training methods in the region of near-homogeneous situations; on the other hand, it implies that locally fair training may fail when data distributions are highly different. Thus, together with Theorem 2, we provide a fundamental understanding of the first question asked in Section 3. Furthermore, the proper group-based fairness metrics provide the possibility to calculate the global fairness value using information from local clients. In the next section, we will utilize this observation and propose a globally fair training algorithm, which answers the second question in Section 3.

## 5 Beyond local fairness

Recall the ultimate goal is to obtain a fair and accurate global model. In the centralized setting, it is standard to minimize the empirical loss with fairness regularization (Bellamy et al., 2019; Berk et al., 2017) as follows:

$$\min_\theta L(\theta) := \sum_{k=1}^K \frac{n_k}{n} L_k(\theta) + \lambda J(F(f(\cdot; \theta); \widehat{\mathcal{D}})), \tag{2}$$

where $J(\cdot)$ is a regularization function and $\lambda$ is a penalty parameter for the fairness term. Without the fairness regularization, Eq. 2 is reduced to Eq. 1, where the gradient of the global objective function can be calculated or estimated by aggregating the gradients of local objective functions $L_k$. FedAvg (McMahan et al., 2017) is inspired by this observation, which performs the gradient descent algorithm on each local client and then aggregates local models. Thus, to generalize federated learning algorithms to fairness-regularized objective Eq. 2, the challenge is how to obtain the gradient of global fairness using summary statistics from local clients. However, as we showed in Corollary 3, the global fairness value cannot be simply represented by local fairness in general.

Fortunately, the next theorem shows that for a proper group-based fairness metric $F$, the gradient of Eq. 2 can be calculated from the gradients of fairness-specific local objectives.

**Theorem 8.** *Let $b = \sum_k w_k b_k$, $d = \sum_k w_k d_k$, and $F_k = a_k/b - c_k/d$. For a proper group-based fairness metric $F$, we have*

$$\widetilde{F} = \sum_{k=1}^K w_k F_k, \quad F(f, \widehat{\mathcal{D}}) = |\widetilde{F}|, \ \nabla_\theta J(F(f, \widehat{\mathcal{D}})) = C_\theta \sum_{k=1}^K w_k \nabla_\theta F_k,$$

*where $C_\theta = \text{sign}(\widetilde{F}) \nabla_F J(F(f, \widehat{\mathcal{D}}))$ is a constant of $F_k$'s.*

Theorem 8 indicates that we can apply the gradient descent algorithm to minimize Eq. 2, similar to the centralized setting. Specifically, at each round $t$, the local client should optimize the following fairness-augmented objective

$$\min_\theta L_k(\theta) + \lambda C_{\theta^{t-1}} F_k(\theta), \tag{3}$$

then the aggregation of local models will give the correct gradient descent update of the global objective function.

Motivated by Theorem 8, we propose a globally fair training method named FedGFT and summarize it in Algorithm 1. We note that FedGFT can incorporate most existing FL algorithms. While the aggregation

---

**Algorithm 1** (FedGFT) Federated learning with globally fair training

---

**Input:** Communication rounds $T$, learning rate $\eta$, local training epochs $E$, batch size $B$, penalty parameter $\lambda$.

**System executes:**
- Initialize the global model parameters $\theta^0$
- **for** each communication round $t = 1, 2, \ldots T$ **do**
  - Sample a subset $S_t \subseteq \{1, \ldots, K\}$
  - Update the constant $C_{\theta^{t-1}} \leftarrow$ **ConstUpdate**$(\theta^{t-1})$
  - **for** each client $k \in S_t$ **in parallel do**
    - Receive the model parameters $\theta_k^{t,0} = \theta^{t-1}$ from the server
    - $\theta_k^{t,E} \leftarrow$ **ClientUpdate**$(\theta_k^{t,0}, C_{\theta^{t-1}})$
  - **end**
  - Server update global model $\theta^t$ by aggregating $\theta_k^{t,E}, k \in S_t$, using any FL algorithm
- **end**
- Return the final global model $f(\cdot; \theta^T)$

**ConstUpdate** $(\theta^t)$:
- **for** each client $k \in S_t$ **in parallel do**
  - Calculate $F_k(\theta^t)$
- **end**
- $\widetilde{F} \leftarrow \sum_{k \in S_t} w_k F_k$
- Return $\nabla_F J(|\widetilde{F}|) \operatorname{sign}(\widetilde{F})$

**ClientUpdate** $(\theta_k^{t,0}, C_{\theta^{t-1}})$:
- **for** each local epoch $e$ from 1 to $E$ **do**
  - **for** each batch $b$ from 1 to $B$ **do**
    - Update model parameters by any FL algorithm with local objective Eq. 3
  - **end**
- **end**
- Return $\theta_k^{t,E}$

---

method on the server side and the optimization tool on the client side remain the same, FedGFT adapts the local objective function to the fairness regularization. Moreover, FedGFT also applies to the situation where clients are purely from one group (for example a client with all points from Group A, and another client with all points from group B), which is not allowed for LFT.

*Remark* 9. Many fairness metrics are not differentiable. Taking SP for example, $a_k = \mathbb{P}(\widehat{Y} = 1, A = 1) = \sum_{i=1}^{n_k} \mathbf{1}_{f(\mathbf{X}_k^{(i)}) > 0.5} \mathbf{1}_{A_k^{(i)} = 0}$ is not differentiable. To employ gradient-based optimization method, a common strategy is using a surrogate, such as the softmax score $\sum_{i=1}^{n_k} f(\mathbf{X}_k^{(i)}) \mathbf{1}_{A_k^{(i)} = 0}$.

Furthermore, we prove that FedGFT will converge to a stationary point when we use gradient-based optimization tools. The complete statement and proof are included in the supplementary document.

**Theorem 10** (Covergence analysis)**.** *Suppose the local clients apply one-step stochastic gradient descent to optimize Eq. 3, and the global server updates the global model by averaging a random subset of local models. Let $\theta^t$ be the parameter of the global model at round $t$. Under mild assumptions, for a step-size sequence $\{\eta_t, t = 0, \ldots, T-1\}$, we have*

$$\min_{t=0,\ldots,T-1} \mathbb{E}(\|\nabla L(\theta^t)\|^2) \leq C \left( \frac{1}{\sum_{t=0}^{T-1} \eta_t} + \frac{\sum_{t=0}^{T-1} \eta_t^2}{\sum_{t=0}^{T-1} \eta_t} \right),$$

*where $C$ is a constant independent of $T$ and $\eta_t$.*

**Corollary 11.** *The choice of $\eta_t = O(1/t), t \geq 1$ yields $\min_{t=0,\ldots,T-1} \mathbb{E}(\|\nabla L(\theta^t)\|^2) \leq O(1/\log(T))$, where $O$ is the big-O notation. The choice of $\eta_t = O(t^{-1/2})$ yields $\min_{t=0,\ldots,T-1} \mathbb{E}(\|\nabla L(\theta^t)\|^2) \leq O(\log(T)T^{-1/2})$. Furthermore, if the gradient descent algorithm is used for optimization instead of stochastic gradient descent, then choosing $\eta_t = \eta_0$ yields a faster rate: $\min_{t=0,\ldots,T-1} \mathbb{E}(\|\nabla L(\theta^t)\|^2) \leq O(T^{-1})$.*

*Remark* 12 (Analysis of additional computation and communication costs of FedGFT.). Compared to classical federated learning algorithm, FedGFT involves an additional step of updating a fairness-related constant at the beginning of each federated round, as shown in Algorithm 1. In particular, this constant update step calculates $F_k = a_k/b - c_k/d$ for each client $k$, where $a_k$ and $c_k$ are specified by the fairness criterion and the local model, while $b$ and $d$ are two universal constants. Moreover, $a_k$ and $c_k$ can be obtained via forward propagating the local model on the local dataset for once. As a result, the additional computation cost is one round of forward propagation step, while the computation cost for a typical federated learning algorithm at each federated round is $E$ rounds of forward and backward propagation steps, where $E$ is the number of epochs.

As for the communication cost, our originally proposed Algorithm 1 has an addition round to synchronize the constant, as pointed out by the reviewer. Nevertheless, this communication round only transmits a single scalar, therefore the cost is much smaller than transmitting model parameters. Furthermore, a possible way to eliminate this additional communication round is synchronizing the constant at the same round as model parameters. Specifically, the local clients can calculate an approximation of $F_k$ at the end of their local training round using the local model, and pass it to the global model. In this way, we do not need a separate round of constant update, at the cost of inaccurate calculation of $F_k$.

*Remark* 13 (Extension to multi-class cases). For a multi-class group variable, say, $A \in \{1, \ldots, C\}$ for some positive integer $C \geq 2$, one possible direction to generalize the definition of group-based fairness metrics is considering

$$F(f, \mathcal{D}) = \sum_{1 \leq i < j \leq C} \left| \frac{a_i(f, \mathcal{D})}{b_i(f, \mathcal{D})} - \frac{a_j(f, \mathcal{D})}{b_j(f, \mathcal{D})} \right|,$$

where $a_i(f, \mathcal{D})$ and $b_i(f, \mathcal{D})$ are some expectations on the event $\{A = i\}$. Accordingly, a proper group-based fairness metric is one such that $b_i$'s are degenerated with respect to $f$. We believe that by using the same techniques as the current work, we can derive similar results as Theorem 2, 4, and 7.

If there is more than one sensitive attribute, say, $A_s \in \{1, \ldots, C_s\}$ for $s = 1, \ldots, S$, we can analogously consider

$$F(f, \mathcal{D}) = \sum_{1 \leq s \leq S} \sum_{1 \leq i < j \leq C_S} \left| \frac{a_i^s(f, \mathcal{D})}{b_i^s(f, \mathcal{D})} - \frac{a_j^s(f, \mathcal{D})}{b_j^s(f, \mathcal{D})} \right|.$$

There are also other ways to formulate the fairness metric for multi-class attributes, such as the variance of ratios $a_i/b_i$, instead of pairwise differences. Nevertheless, the proposed FedGFT can be extended and applied to those scenarios as long as $a/b$ can be obtained using local statistics.

## 6 Experiments

### 6.1 Performance under Different Data Heterogeneity levels

**Datasets.** We use the following three datasets in this subsection.

1. Adult dataset (Dua & Graff, 2017). This dataset contains the income level and demographic attributes of 48842 people. We train a logistic regression model to predict a binary response 'Income' (high or low) with 14 continuous and categorical predictors. The predictor 'Race' (white or non-white) is considered the sensitive group.

2. COMPAS dataset (Angwin et al., 2016). This dataset includes ten demographic attributes of 6172 criminal defendants and whether they recidivate in two years. A logistic model is trained to predict recidivism, and gender is the sensitive variable.

3. CelebA dataset (Liu et al., 2015). This dataset contains $202,599$ face images, and each image has 40 binary attributes. We train a ResNet18 model (He et al., 2016) targeting at classifying the 'Smiling' attribute (yes or no), and take 'Male' (yes or no) as the sensitive attribute. To speed up the training

Table 3: Hyper-parameters used in our experiments.

| Dataset | Adult | COMPAS | CelebA |
|---|---|---|---|
| Architecture | Linear | Linear | ResNet18 |
| Number of clients | 10 | 10 | 10 |
| Communication round | 50 | 50 | 50 |
| Batch size | 256 | 256 | 64 |
| Epoch | 1 | 3 | 1 |
| Optimizer | ADAM | ADAM | ADAM |
| Learning rate | 0.002 | 0.01 | 0.001 |
| Scheduler | N/A | N/A | MultistepLR |
| Weight decay | N/A | N/A | 0.1 |

process, we randomly select $10,000$ images for training and $6,000$ for testing in each independent experiment.

**Setup.** There are six methods in comparison: baseline method 'FedAvg' (McMahan et al., 2017), state-of-the-art fairness-aware FL 'FairFed' (Ezzeldin et al., 2021) and 'FedFB' (Zeng et al., 2021), LFT methods locally reweighing 'LRW' (Abay et al., 2020) and 'FedLFT', and our proposed globally fair method 'FedGFT'. For each dataset, we first randomly split the original dataset into three parts, training, validation, and test dataset. The training dataset is further split into disjoint subsets, serving as local datasets of clients. The steps of dividing the training dataset are detailed as follows. First, we generate the proportion of each combination of the group variable $A$ and response variable $Y$ for each client from a Dirichlet distribution $\text{Dir}(\alpha)$. A larger $\alpha$ implies more homogeneous clients. Then, we randomly assign the corresponding proportion of data points to each client. Throughout this section, $\alpha$ takes values in $0.5, 5$, and $100$. After training is done, the test accuracy (using zero-one loss) and fairness metric of the global model is calculated on the test dataset. Note that we conduct the experiments using both SP and EOP as the fairness metric, respectively. For each combination of dataset, learning method, and $\alpha$, we run 20 independent experiments.

The default hyper-parameters are listed in Table 3. The penalty parameter for 'FairFed' is chosen from $\{0.1, 1, 10\}$ with cross-validation, and for 'FedGFT' is chosen from $\{10, 20, 50\}$. Also, the regularization function used by FedGFT is $J(x) = x^2$.

**Results.** Experiment results are summarized in Table 4. We can see that FedGFT has the smallest bias among almost all situations, while the accuracy drop by using FedGFT is negligible. Notably, 'FedLFT' and 'FedFB' achieve similar fairness performance as 'FedGFT' but experience markedly degraded accuracy. 'LRW' is worse than 'FedGFT' except for EOP in the CelebA dataset. While the accuracy of FedGFT is within two standard errors compared to the best method, FedGFT significantly decreases the bias even in the highly heterogeneous case.

**Discussion on the choice of hyper-parameters for FedGFT.** The three datasets used in this paper differ in learning difficulty, which influences the choice of proper hyper-parameters. We also conduct the experiments that vary each hyper-parameter with communication round $T = 50$, which ensures that the model achieves convergence at the end of training. On the COMPAS dataset, we observe that the learning rate and number of epochs will jointly determine the performance of FedGFT. When the learning rate and number of epochs is increased, it will first accelerate model convergence (less communication round), since there are more local updating steps in each round and the step size is increased. However, when the learning rate or the number of epochs is too large, it will impede model training instead. In particular, a large learning rate will cause the local model to oscillate around the minimizer and hence hard to converge, which is a known issue of the classical neural network training using an optimizer with a fixed learning rate. On the other hand, a large number of epochs can amplify the difference among local models at each communication

Table 4: The average accuracy and bias (standard error in parentheses) on three datasets, under three heterogeneity levels and two fairness metrics. The proposed method is marked by †.

| | Dataset | Adult | | | COMPAS | | | CelebA | | |
|---|---|---|---|---|---|---|---|---|---|---|
| $\alpha$ | Method | Acc (↑) | SP (↓) | EOP (↓) | Acc(↑) | SP (↓) | EOP (↓) | Acc (↑) | SP (↓) | EOP (↓) |
| 0.5 | FedAvg | 81.8 (0.4) | 5.7 (1.0) | 18.3 (3.3) | **64.2 (0.5)** | 12.8 (1.8) | 15.2 (2.0) | 91.4 (0.7) | 14.7 (3.2) | 7.5 (2.7) |
| | LRW | 81.1 (0.5) | 1.9 (0.4) | 11.8 (0.5) | 60.9 (1.3) | 5.1 (0.8) | 6.4 (1.4) | **91.9 (0.4)** | 13.7 (1.0) | **1.3 (0.7)** |
| | FairFed | 80.8 (0.4) | 2.0 (0.4) | 10.8 (0.5) | 61.2 (1.0) | 5.2 (0.7) | 3.6 (0.6) | 91.7 (0.4) | 13.8 (3.0) | 7.7 (2.9) |
| | FedLFT | 79.3 (0.3) | 3.4 (0.7) | 2.1 (0.4) | 61.6 (0.9) | 2.3 (0.2) | 4.0 (0.4) | 86.2 (7.9) | **6.4 (3.3)** | 3.7 (2.5) |
| | FedGFT† | **81.8 (0.4)** | **0.8 (0.1)** | **1.3 (0.3)** | 63.6 (0.6) | **0.9 (0.1)** | **1.3 (0.2)** | 88.9 (3.6) | 8.1 (5.1) | 1.7 (2.3) |
| 5 | FedAvg | 83.3 (0.1) | 6.2 (0.2) | 3.6 (0.4) | **66.7 (0.2)** | 12.4 (0.5) | 12.0 (0.5) | **92.0 (0.3)** | 13.6 (0.4) | 5.8 (0.4) |
| | LRW | **83.4 (0.1)** | 2.2 (0.1) | 10.2 (0.1) | 66.3 (0.3) | 3.6 (0.2) | 3.8 (0.3) | 91.8 (0.3) | 13.8 (0.2) | **0.4 (0.2)** |
| | FairFed | 83.4 (0.1) | 2.0 (0.1) | 10.3 (0.1) | 65.9 (0.3) | 3.8 (0.4) | 3.6 (0.4) | 91.9 (0.3) | 13.8 (0.3) | 6.0 (0.5) |
| | FedLFT | 82.4 (0.2) | 2.4 (0.2) | 2.6 (0.2) | 65.3 (0.3) | 1.5 (0.1) | 2.1 (0.2) | 91.7 (0.4) | 6.0 (1.0) | 5.3 (1.3) |
| | FedGFT† | 83.1 (0.1) | **0.5 (0.0)** | **0.5 (0.1)** | 66.1 (0.2) | **0.5 (0.0)** | **0.7 (0.0)** | 90.7 (0.5) | **4.9 (1.6)** | 0.7 (0.4) |
| 100 | FedAvg | 83.3 (0.1) | 6.9 (0.1) | 2.3 (0.1) | **67.1 (0.2)** | 13.4 (0.2) | 13.5 (0.2) | **92.0 (0.3)** | 13.6 (0.2) | 5.7 (0.3) |
| | LRW | **83.4 (0.1)** | 2.4 (0.0) | 10.0 (0.1) | 65.9 (0.2) | 2.9 (0.1) | 2.8 (0.1) | 91.9 (0.3) | 13.8 (0.1) | **0.2 (0.1)** |
| | FairFed | 83.1 (0.1) | 2.5 (0.1) | 9.8 (0.1) | 65.9 (0.2) | 2.9 (0.2) | 2.7 (0.2) | 91.4 (0.4) | 13.7 (0.1) | 5.8 (0.3) |
| | FedLFT | 83.2 (0.1) | 2.0 (0.1) | 2.9 (0.1) | 65.8 (0.2) | 1.4 (0.1) | 2.1 (0.2) | 91.4 (0.4) | **6.0 (0.4)** | 5.4 (1.2) |
| | FedGFT† | 83.3 (0.1) | **0.5 (0.0)** | **0.6 (0.1)** | 66.3 (0.2) | **0.5 (0.0)** | **0.5 (0.0)** | 90.8 (0.7) | 6.4 (3.3) | 0.3 (0.3) |
| | FedFB | 76.6 (0.3) | 1.6 (1.5) | NA | 54.2 (0.0) | 0.1 (0.0) | NA | NA | NA | NA |

round, causing slow convergence for the global model. Nevertheless, according to the above analysis, we may accommodate the choice of hyper-parameter by using an optimizer with shrinking learning rate and an objective that can address data heterogeneity. Moreover, experimental results with different combination of hyper-parameters consistently show that FedGFT outperforms other methods. Lastly, it is recommended to choose those hyper-parameters through cross-validation.

**Discussion on the required conditions of FedGFT.** In our originally proposed FedGFT, there are two assumptions: (1) the fairness metric is proper, (2) all clients are involved in each federated round. Here, a proper group-based fairness metric plays an essential role for FedGFT, while the second condition can be relaxed. Specifically, for cross-device federated learning, we can apply FedGFT as long as clients are independently sampled in each round. Recall that our ultimate goal is optimizing the penalized objective Eq. (2), and FedGFT shows that its gradient can be written as the linear combination of gradients from local clients. When clients are sampled independently, then we can replace the whole gradient with its stochastic counterpart, namely the combination of gradients from those independently sampled clients. This is akin to the relationship between gradient descent algorithm and stochastic gradient descent algorithm.

## 6.2 Performance under the Extreme Case of Data Heterogeneity

In this subsection, we performed experiments under the scenario where each client's data are purely from one group. In this case, local fairness is not well-defined. Therefore, locally fair training is not applicable, and our proposed FedGFT is preferred instead. With other settings the same as above subsection, we conduct experiments on the COMPAS dataset to corroborate our algorithm's effectiveness. The results are summarized in Table 5. From the results, FedGFT still mitigates the bias compared to FedAvg, though the accuracy-fairness trade-off is worse than the situation where the clients have data from both groups.

## 6.3 Performance under Different Client Heterogeneity Levels

In this subsection, we investigate the learning algorithm's performance given the following two scenarios: 1. there are many more clients. 2. The number of data points owned by each client is highly unequal. We conduct experiments on the COMPAS dataset and compare all six methods except FedFB.

For the first scenario, with other setting the same as in Subsection 6.1, the number of clients is 100. That is, each client has tens of data points on average. The results are reported in Table 6. Compared to Table 4

Table 5: The average accuracy and bias (standard error in parentheses) on the COMPAS dataset under two fairness metrics. Pure group represents the situation where clients are purely from one group; mixed group represents the case where clients have data from both groups; and $\lambda$ is the penalty parameter.

| Method | Acc | SP | EOP |
|---|---|---|---|
| FedAvg (Mixed group) | **66.7 (0.2)** | 12.4 (0.5) | 12.0 (0.5) |
| FedGFT (Mixed group, $\lambda = 20$) | 66.1 (0.2) | **0.5 (0.0)** | 0.7 (0.0) |
| FedAvg (Pure Group) | 66.6 (0.2) | 6.9 (0.5) | 6.2 (0.4) |
| FedGFT (Pure group, $\lambda = 10$) | 65.5 (0.3) | 2.4 (0.3) | 2.2 (0.2) |
| FedGFT (Pure group, $\lambda = 20$ ) | 64.4 (0.4) | 1.7 (0.2) | 1.4 (0.1) |
| FedGFT (Pure group, $\lambda = 100$ ) | 59.0 (0.8) | 1.0 (0.1) | **0.5 (0.1)** |

Table 6: The average accuracy and bias (standard error in parentheses) on COMPAS datasets when there are 100 clients.

| $\alpha$ | Method | Acc ($\uparrow$) | SP ($\downarrow$) | EOP ($\downarrow$) |
|---|---|---|---|---|
| 0.5 | FedAvg | 66.0 (0.2) | 9.0 (0.8) | 9.3 (2.0) |
| | LRW | **66.9 (0.4)** | 9.6 (0.5) | 6.5 (1.3) |
| | FairFed | 61.9 (1.7) | 5.3 (0.5) | 3.9 (0.8) |
| | FedLFT | 61.6 (0.9) | 0.9 (0.1) | 1.7 (0.3) |
| | FedGFT | 62.2 (1.6) | **0.8 (0.1)** | **1.1 (0.2)** |
| 5.0 | FedAvg | **67.4 (0.3)** | 12.9 (0.2) | 13.0 (0.2) |
| | LRW | 66.3 (0.3) | 3.7 (0.2) | 3.5 (0.2) |
| | FairFed | 66.7 (0.3) | 3.6 (0.2) | 3.0 (0.2) |
| | FedLFT | 66.2 (0.4) | 1.3 (0.1) | 1.0 (0.1) |
| | FedGFT | 66.0 (0.3) | **0.5 (0.0)** | **0.7 (0.0)** |
| 100.0 | FedAvg | **67.6 (0.4)** | 14.2 (0.1) | 14.1 (0.1) |
| | LRW | 66.9 (0.2) | 2.5 (0.1) | 2.6 (0.1) |
| | FairFed | 66.4 (0.3) | 2.3 (0.1) | 2.3 (0.2) |
| | FedLFT | 66.5 (0.2) | 1.1 (0.1) | 1.1 (0.1) |
| | FedGFT | 66.8 (0.2) | **0.5 (0.0)** | **0.5 (0.0)** |

where only 10 clients are involved, FedGFT is still the algorithm with the smallest bias. Moreover, the accuracy drop by FedGFT is similar or smaller to other fairness-aware algorithms.

Next, we consider the scenario where each client can have different number of data points. With other setting the same as in Subsection 6.1, we modify the data spliting step as follows. First, we generate the proportion of data points for each client from a Dirichlet distribution $\text{Dir}(\alpha)$. Then, for each combination of sensitive attribute $A$ and response $Y$, we randomly assign the corresponding proportion of data points to each client. It means that the data is completely homogeneous to each client, which is the most favorable scenario for locally fair training methods. The parameter $\alpha$ takes values in $0.5, 5$, and $100$, and a higher $\alpha$ indicates more evenly distributed numbers of data points. For example, the median ratio of the largest and smallest number of data points is around 400 when $\alpha = 0.5$, while this ratio is 1.3 for $\alpha = 100$.

The results of clients with varying numbers of data points are reported in Table 7. Clearly, FedGFT has the smallest bias even in the homogeneous setting, with a comparable accuracy to FedAvg. The results, together with the experiments under different data heterogeneity levels, demonstrate that FedGFT is a robust and effective fairness-aware federated learning algorithm.

Table 7: The average accuracy and bias (standard error in parentheses) on COMPAS datasets when each client's number of data points can vary.

| $\alpha$ | Method | Acc ($\uparrow$) | SP ($\downarrow$) | EOP ($\downarrow$) |
|---|---|---|---|---|
| 0.5 | FedAvg | 65.7 (0.2) | 22.4 (0.9) | 23.3 (0.8) |
| | LRW | **66.4 (0.3)** | 3.6 (0.7) | 2.6 (0.3) |
| | FairFed | 65.8 (0.3) | 2.2 (0.6) | 1.8 (0.4) |
| | FedLFT | 64.8 (0.4) | 1.4 (0.4) | 3.8 (0.8) |
| | FedGFT | 65.4 (0.3) | **0.3 (0.0)** | **0.4 (0.1)** |
| 5.0 | FedAvg | 66.3 (0.3) | 27.1 (0.2) | 26.4 (0.4) |
| | LRW | **67.0 (0.2)** | 5.4 (0.4) | 3.1 (0.3) |
| | FairFed | 65.9 (0.4) | 4.3 (0.8) | 0.9 (0.1) |
| | FedLFT | 65.3 (0.3) | 0.9 (0.1) | 1.4 (0.3) |
| | FedGFT | 65.3 (0.2) | **0.3 (0.0)** | **0.4 (0.0)** |
| 100.0 | FedAvg | 65.5 (0.3) | 26.0 (0.1) | 26.5 (0.1) |
| | LRW | **66.4 (0.3)** | 3.4 (0.1) | 3.3 (0.2) |
| | FairFed | 65.6 (0.3) | 2.7 (0.4) | 2.5 (0.3) |
| | FedLFT | 65.9 (0.3) | 1.0 (0.2) | 0.7 (0.1) |
| | FedGFT | 65.4 (0.2) | **0.3 (0.0)** | **0.3 (0.0)** |

## 7 Conclusion

In this work, we proved that the fairness of the global model in federated learning is upper-bounded by the fairness of local models and the data heterogeneity level for proper group-based fairness metrics, thus providing theoretical support for locally fair training methods. Nevertheless, locally fair training may fail in highly heterogeneous cases. We also proposed a globally fair training method called FedGFT for proper group-based fairness metrics, which directly minimizes the fairness-penalized empirical loss of the global model and can be easily incorporated with existing FL algorithms. Experiments on three real-world datasets showed that the proposed method can significantly reduce the model bias while retaining a similar prediction accuracy compared to the baseline.

**Limitations** There are several problems not fully addressed and will be interesting future work. For example, the calibration is not a proper fairness metric, thus, how to generalize our results to calibration, or more generally, group-based metrics, is of interest. It is also worth thinking about how to generalize the proposed method regarding multiple group variables. Moreover, an examination of the proposed method on large-scale datasets is desired. For instance, Ding et al. (2021) proposed to retire the Adult dataset and use a suite of larger datasets derived from US Census surveys for fair machine learning research.

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

## A Fairness metrics

In this appendix section, we validate that three fairness metrics (SP, EOP, and Calibration) satisfy our Definition 1. Furthermore, SP and EOP are proper group-based fairness metric Definition 5.

**Statistical Parity.** Recall that $F(f, \mathcal{D}) = |\mathbb{P}(\widehat{Y} = 1|A = 0) - \mathbb{P}(\widehat{Y} = 1|A = 1)|$. By Bayes' Theorem, we have

$$\mathbb{P}(\widehat{Y} = 1|A = 0) = \frac{\mathbb{P}(\widehat{Y} = 1, A = 0)}{\mathbb{P}(A = 0)}.$$

Therefore, let $a(f, \mathcal{D}) = \mathbb{P}(\widehat{Y} = 1, A = 0)$, $b(f, \mathcal{D}) = \mathbb{P}(A = 0)$, then $\mathbb{P}(\widehat{Y} = 1|A = 0) = a(f, \mathcal{D})/b(f, \mathcal{D})$. Similarly, we have $\mathbb{P}(\widehat{Y} = 1|A = 1) = c(f, \mathcal{D})/d(f, \mathcal{D})$, where $c(f, \mathcal{D}) = \mathbb{P}(\widehat{Y} = 1, A = 1)$, $d(f, \mathcal{D}) = \mathbb{P}(A = 1)$. Thus, SP is a group-based fairness metric. Furthermore, it is clear that $b(f, \mathcal{D})$ and $d(f, \mathcal{D})$ are independent of $f$, hence SP is also a proper group-based fairness metric.

**Equal Opportunity.** For EOP, $F(f, \mathcal{D}) = |\mathbb{P}(\widehat{Y} = 1|A = 0, Y = 1) - \mathbb{P}(\widehat{Y} = 1|A = 1, Y = 1)|$. Using Bayes' Theorem again, we know

$$\mathbb{P}(\widehat{Y} = 1|A = 0, Y = 1) = \frac{\mathbb{P}(\widehat{Y} = 1, A = 0, Y = 1)}{\mathbb{P}(A = 0, Y = 1)},$$

$$\mathbb{P}(\widehat{Y} = 1|A = 1, Y = 1) = \frac{\mathbb{P}(\widehat{Y} = 1, A = 1, Y = 1)}{\mathbb{P}(A = 1, Y = 1)},$$

which aligns with Table 1.

**Well-Calibration.** In this case, $F(f, \mathcal{D}) = |\mathbb{P}(Y = 1|A = 0, \widehat{Y} = 1) - \mathbb{P}(Y = 1|A = 1, \widehat{Y} = 1)|$, and

$$\mathbb{P}(Y = 1|A = 0, \widehat{Y} = 1) = \frac{\mathbb{P}(Y = 1, A = 0, \widehat{Y} = 1)}{\mathbb{P}(A = 0, \widehat{Y} = 1)},$$

$$\mathbb{P}(Y = 1|A = 1, \widehat{Y} = 1) = \frac{\mathbb{P}(Y = 1, A = 1, \widehat{Y} = 1)}{\mathbb{P}(A = 1, \widehat{Y} = 1)}.$$

We note that for calibration, $b(f, \mathcal{D}) = \mathbb{P}(A = 0, \widehat{Y} = 1)$ and $d(f, \mathcal{D}) = \mathbb{P}(A = 1, \widehat{Y} = 1)$, which are functions of both function $f$ and distribution $\mathcal{D}$.

## B Missing Proofs in Section 4

**Proof of Theorem 2.** We first prove that fair local models do not imply a fair global model. Let $g_k$ and $g$ be the abbreviations of $g(f, \widehat{\mathcal{D}}_k)$ and $g(f, \widehat{\mathcal{D}})$ for $g \in \{a, b, c, d\}$ (see Definition 1), respectively. Note that the summation in $g\left(f, \sum_k w_k D_k\right)$ can be taken out because $g$ is an expectation by Definition 1. For example, $a(f, D)$ can be written as $\int_{(X,Y)\sim D} h(f, X, Y) dX dY$ for some function $h$. Specifically, $a(f, D) = \int_{(X,Y)\sim D} 1_{\{f(X)>0.5, A=0\}} dX dY$ for statistical disparity, where $1_{\{\cdot\}}$ is an indicator function. Now, the basic

property of integral gives that

$$a\left(f,\sum_k w_k D_k\right) = \int_{(X,Y)\sim\sum_k w_k D_k} h(f,X,Y)dXdY$$

$$= \sum_k w_k \int_{(X,Y)\sim D_k} h(f,X,Y)dXdY = \sum_k w_k a(f,D_k).$$

As a result, we only need to show that there exist a $f$ and data distributions $\widehat{\mathcal{D}}_k$'s such that

$$F(f,\widehat{\mathcal{D}}) = F(f,\sum_k w_k\widehat{\mathcal{D}}_k) = \left|\frac{\sum_k w_k a_k}{\sum_k w_k b_k} - \frac{\sum_k w_k c_k}{\sum_k w_k d_k}\right| \geq C, \tag{4}$$

where $w_k = n_k/n$.

Note that $w_k$ can take an arbitrary value in $[0,1]$ as long as $\widehat{\mathcal{D}}_k$'s are properly chosen. Furthermore, according to Definition 1, $g_k$'s are arbitrarily manipulable as well. Next, we will construct $g_k$'s and $w_k$'s that satisfy Eq. 4. In particular, we consider quantities with $a_1/b_1 = c_1/d_1 = 1$, $w_1 = (1+C)/2$, and $a_k/b_k = c_k/d_k = 0$ and $w_k = (1-w_1)/(K-1)$ for $k = 2,\ldots,K$. By simple calculation, when $b_1, d_2, \ldots, d_K$ converge to one and $d_1, b_2, \ldots, b_K$ converge to zero, $F(f,\widehat{\mathcal{D}})$ converges to $w_1$, which is larger than $C$. It immediately implies that there exists a proper choice satisfying Eq. 4.

As for the converse result, the following choice suffices:

$$a_{2l}/b_{2l} = C, a_{2l+1}/b_{2l+1} = 0,$$
$$c_{2l}/d_{2l} = 0, c_{2l+1}/d_{2l+1} = C,$$
$$w_{2l} = \frac{\lfloor (K+1)/2 \rfloor}{2\lfloor K/2 \rfloor \lfloor (K+1)/2 \rfloor},$$
$$w_{2l+1} = \frac{\lfloor K/2 \rfloor}{2\lfloor K/2 \rfloor \lfloor (K+1)/2 \rfloor}, l = 0,\ldots,\lfloor K/2 \rfloor,$$

where $\lfloor x \rfloor$ means the floor of a number $x$.

**Proof of Corollary 3.** If the claim is false, then there exists a sequence of constants $\{v_k, k = 1,\ldots,K\}$, such that $F(f,\widehat{\mathcal{D}}) = \sum_{k=1}^K v_k F(f,\widehat{\mathcal{D}}_k)$ always holds. Now, evoking Theorem 2, we know it is possible that $F(f,\widehat{\mathcal{D}}) > 0$ with $F(f,\widehat{\mathcal{D}}_k) = 0$ for all $k$, which is a contradiction, and thus concludes the proof.

**Proof of Theorem 4.** Recall that $F(f,\mathcal{D}_k) = |a_k/b_k - c_k/d_k|$. From Eq. 4, we know that

$$F(f,\mathcal{D}) = 0 \iff \frac{\sum_k w_k a_k}{\sum_k w_k b_k} = \frac{\sum_k w_k c_k}{\sum_k w_k d_k}.$$

Multiplying $(\sum_k w_k b_k)(\sum_k w_k d_k)$ on the both hand sides and rearranging the above equation yields that $\boldsymbol{w}^\mathrm{T} M \boldsymbol{w} = 0$, where $\boldsymbol{w} = (w_1,\ldots,w_K)^\mathrm{T}$ and $M \in \mathbb{R}^{K\times K}$ is a matrix with $(i,j)$-th element $M_{ij} = a_i d_j - b_i c_j$. Thus, $\boldsymbol{w}^\mathrm{T} M \boldsymbol{w} = 0$ for any $\boldsymbol{w}$ is equivalent to that $a_i d_j - b_i c_j = 0$ for all $1 \leq i, j \leq K$, which completes the proof.

**Proof of Theorem 7.** The local fairness condition $F(f,\widehat{\mathcal{D}}_k) \leq \alpha$ gives that $|a_k/b_k - c_k/d_k| \leq \alpha$, thus $a_k \leq (\alpha + c_k/d_k)b_k$, and we have

$$\frac{\sum_k w_k a_k}{\sum_k w_k b_k} - \frac{\sum_k w_k c_k}{\sum_k w_k d_k} \leq \alpha + \frac{\sum_k w_k c_k b_k/d_k}{\sum_k w_k b_k} - \frac{\sum_k w_k c_k}{\sum_k w_k d_k} \leq \alpha + \frac{\sum_k w_k c_k}{\sum_k w_k d_k}\left(\frac{d}{b}\frac{b_k}{d_k} - 1\right) \leq \alpha + \beta.$$

The last step is due to $c_k/d_k \leq 1$ and the definition of data heterogeneity coefficient. Similarly, we have $a_k \geq (c_k/d_k - \alpha)b_k$ and

$$\frac{\sum_k w_k a_k}{\sum_k w_k b_k} - \frac{\sum_k w_k c_k}{\sum_k w_k d_k} \geq \frac{\sum_k w_k c_k b_k/d_k}{\sum_k w_k b_k} - \alpha - \frac{\sum_k w_k c_k}{\sum_k w_k d_k}$$

$$\geq -\alpha + \frac{\sum_k w_k c_k}{\sum_k w_k d_k}\left(\frac{d}{b}\frac{b_k}{d_k} - 1\right) \geq -(\alpha + \beta),$$

which concludes the proof.

## C  Missing Proofs in Section 5

### C.1  Proof of Theorem 8.

For a proper group-based fairness metric $F$, we have

$$F(f, \widehat{\mathcal{D}}_k) = \left| \frac{a_k}{b_k} - \frac{c_k}{d_k} \right|,$$

$$F(f, \widehat{\mathcal{D}}) = \left| \frac{\sum_k w_k a_k}{\sum_k w_k b_k} - \frac{\sum_k w_k c_k}{\sum_k w_k d_k} \right|,$$

where $a_k$ and $c_k$ are functions of $f$ and $\widehat{\mathcal{D}}_k$, and $b_k$ and $d_k$ are functions of $\widehat{\mathcal{D}}_k$ only. Recall that $b = \sum_k w_k b_k$, $d = \sum_k w_k d_k$, and $F_k = a_k/b - c_k/d$, it is straightforward to verify that

$$F(f, \widehat{\mathcal{D}}) = |\widetilde{F}|, \ \widetilde{F} = \frac{\sum_k w_k a_k}{\sum_k w_k b_k} - \frac{\sum_k w_k c_k}{\sum_k w_k d_k} = \sum_k w_k F_k.$$

Therefore,

$$
\begin{aligned}
\nabla_\theta F(f, \widehat{\mathcal{D}}) &= \operatorname{sign}(\widetilde{F}(f, \widehat{\mathcal{D}})) \left( \frac{\sum_k w_k \nabla_\theta a_k}{\sum_k w_k b_k} - \frac{\sum_k w_k \nabla_\theta c_k}{\sum_k w_k d_k} \right) \\
&= \operatorname{sign}(\widetilde{F}(f, \widehat{\mathcal{D}})) \sum_{k=1}^{K} w_k \left( \frac{\nabla_\theta a_k}{b} - \frac{\nabla_\theta c_k}{d} \right) \\
&= \sum_{k=1}^{K} w_k \operatorname{sign}(\widetilde{F}(f, \widehat{\mathcal{D}})) \nabla_\theta F_k.
\end{aligned}
$$

Finally, by chain rule, we have that

$$\nabla_\theta J(F(f, \widehat{\mathcal{D}})) = \nabla_F J(F(f, \widehat{\mathcal{D}})) \nabla_\theta F(f, \widehat{\mathcal{D}}) = \operatorname{sign}(\widetilde{F}) \nabla_F J(F(f, \widehat{\mathcal{D}})) \sum_{k=1}^{K} w_k \nabla_\theta F_k,$$

which completes the proof.

### C.2  Convergence Analysis

We first restate the problem setup for clarity. Recall the global objective function Eq. equation 2 is

$$\min_\theta L(\theta) = \sum_{k=1}^{K} \frac{n_k}{n} L_k(\theta) + \lambda J(F(f(\cdot; \theta); \widehat{\mathcal{D}})).$$

And the local objective functions are

$$\min_\theta H_k(\theta) := L_k(\theta) + \lambda C_{\theta^{t-1}} F_k(\theta).$$

Next, we state the training procedure with random client selection and stochastic gradient descent optimization. In particular, at the communication round $t$, we have

$$\theta_k^{t+1} = \theta^t - \eta_t g_k(\theta^t \mid \xi), k \in S_t,$$

$$\theta^{t+1} = \frac{1}{K} \sum_{k \in S_t} \theta_k^{t+1},$$

where $g_k(\theta^t \mid \xi)$ is the stochastic gradient of $H_k$, $\xi$ represents the stochastic batches of datasets, $S_t$ is a randomly selected subset of clients with cardinality $M$ (in which client $k$ is selected with probability $n_k/n$), and $\eta_t$ is the step size.

We make the following technical assumptions often used in the optimization literature, e.g., Li et al. (2020c); Wang et al. (2020) and the references therein. For two vectors $u$ and $v$, $\langle u, v \rangle = u^\mathsf{T} v$ is the inner product of $u$ and $v$, and $\|v\| = (v^\mathsf{T} v)^{1/2}$ is the $\ell_2$ norm of $v$. The gradient operator $\nabla$ is with respect to the model parameter $\theta$ throughout this subsection.

**Assumption 14** (Smoothness). The gradients of $L_k$'s and $J$ are $L$-Lipshitz continuous. A function $f(\cdot)$ is $L$-Lipshitz continuous if for any $x, y$ we have

$$\|\nabla f(x) - \nabla f(y)\| \le L\|x - y\|.$$

**Assumption 15** (Unbiasedness). The stochastic gradient is unbiased for all clients, that is, $\mathbb{E}_\xi\{g_k(\theta^t \mid \xi)\} = \nabla H_k(\theta^t)$, for all $k = 1, \ldots, K$.

**Assumption 16** (Bounded variance). The stochastic gradient has a bounded variance for all clients, namely $\mathbb{E}_\xi[\{g_k(\theta^t \mid \xi) - \nabla H_k(\theta^t)\}^2] \le \sigma^2$, for all $k = 1, \ldots, K$.

**Assumption 17** (Bounded dissimilarity). There exist a constant $B \ge 1$ such that for all $\sum_{k=1}^K w_k = 1, w_k \ge 0$, we have

$$\sum_{k=1}^K w_k \|\nabla H_k(\theta)\|^2 \le B^2 \left\| \sum_{k=1}^K w_k \nabla H_k(\theta) \right\|^2.$$

**Assumption 18.** The objective function is lower bounded, $L^* := \inf_\theta L(\theta) > -\infty$.

*Remark* 19. Assumptions 14, 15, and 16 are standard in optimization literature, which ensure that the SGD update produces a sufficiently large decrease in the function value, leading to the convergence. Assumption 17 ensures the convergence with data heterogeneity. Larger $B$ indicate more severe data heterogeneity, and $B = 1$ corresponds to the homogeneous case.

*Remark* 20. If we use GD instead of SGD, then the update of local clients will be

$$\theta_k^{t+1} = \theta_l^{t+1} - \eta_t \nabla H_k(\theta^t), k \in S_t,$$

and the Assumptions 15 and 16 are automatically satisfied with $\sigma = 0$.

Now, we restate and prove Theorem 10 below.

**Theorem 10 (Convergence result)** Under Assumptions 14-18, when the step-size sequence $\{\eta_t, t = 0, \ldots, T-1\}$ satisfies $C_0 \ge \eta_0 \ge \eta_t > 0$, we have

$$\min_{t=0,\ldots,T-1} \mathbb{E}(\|\nabla L(\theta^t)\|^2) \le C\left( \frac{1}{\sum_{t=0}^{T-1} \eta_t} + \frac{\sum_{t=0}^{T-1} \eta_t^2}{\sum_{t=0}^{T-1} \eta_t} \right),$$

where $C_0$ and $C$ are two constants independent of $T$ and $\{\eta_t, t = 0, \ldots, T-1\}$.

**Proof of Theorem 10** We assume $\theta^t$ is fixed for now and denote $\widehat{\theta}^{t+1} := \theta^t - \eta_t \nabla L(\theta^t)$. Note that $\mathbb{E}(\theta^{t+1}) = \widehat{\theta}^{t+1}$ by Theorem 8. By Assumption 14, we have

$$\mathbb{E}\{L(\theta^{t+1})\} \le L(\theta^t) + \mathbb{E}\{\langle \nabla L(\theta^t), \theta^{t+1} - \theta^t \rangle\} + \mathbb{E}\left( \frac{L}{2} \|\theta^{t+1} - \theta^t\|^2 \right)$$

$$\le L(\theta^t) + \langle \nabla L(\theta^t), \widehat{\theta}^{t+1} - \theta^t \rangle L\{\|\widehat{\theta}^{t+1} - \theta^t\|^2 + \mathbb{E}(\|\widehat{\theta}^{t+1} - \theta^{t+1}\|^2)\}$$

$$= L(\theta^t) - \eta_t(1 - L\eta_t)\|\nabla L(\theta^t)\|^2 L\mathbb{E}(\|\widehat{\theta}^{t+1} - \theta^{t+1}\|^2). \tag{5}$$

Since all clients are independent of each other, we have

$$\mathbb{E}(\|\widehat{\theta}^{t+1} - \theta^{t+1}\|^2) = \mathbb{E}_\xi\{\mathbb{E}_{S_t}(\|\widehat{\theta}^{t+1} - \theta^{t+1}\|^2)\} \le \mathbb{E}_\xi\left\{\frac{1}{M}\mathbb{E}_k(\|\theta_k^{t+1} - \widehat{\theta}^{t+1}\|^2)\right\}$$

$$= \frac{\eta_t^2}{M}\mathbb{E}_\xi\left\{\mathbb{E}_k(\|g_k(\theta^t \mid \xi) - \nabla L(\theta^t)\|^2)\right\}$$

$$\le \frac{2\eta_t^2}{M}\left\{\mathbb{E}_k(\|\nabla H_k(\theta^t) - \nabla L(\theta^t)\|^2) + \sigma^2\right\}$$

$$\le \frac{4\eta_t^2}{M}\left\{\mathbb{E}_k(\|\nabla H_k(\theta^t)\|^2) + \|\nabla L(\theta^t)\|^2 + \sigma^2\right\}$$

$$\le \frac{4\eta_t^2}{M}\left\{(B^2 + 1)\|\nabla L(\theta^t)\|^2 + \sigma^2\right\}, \tag{6}$$

where the second and third inequalities is due to triangle inequality and Assumption 16, and the last inequality is due to Assumption 17. Plugging Eqs. equation 6 into Eq. equation 5, we have

$$\mathbb{E}\{L(\theta^{t+1})\} \le L(\theta^t) - \eta_t(1 - L\eta_t)\|\nabla L(\theta^t)\|^2 + 4M^{-1}L\eta_t^2\{(B^2+1)\|\nabla L(\theta^t)\|^2 + \sigma^2\}$$

$$= L(\theta^t) + 4M^{-1}L\eta_t^2\sigma^2 - \eta_t[1 - L\eta_t\{1 + 4M^{-1}L(B^2+1)\}]\|\nabla L(\theta^t)\|^2$$

$$= L(\theta^t) - \eta_t(1 - c_1\eta_t)\|\nabla L(\theta^t)\|^2 + c_2\eta_t^2, \tag{7}$$

where $c_1 = L\{1 + 4M^{-1}L(B^2 + 1)\}$ and $c_2 = 4M^{-1}L\sigma^2$. Next, we take expectation on Eq. equation 7, reorganize and sum it from $t = 0$ to $t = T - 1$, obtaining

$$\sum_{t=0}^{T-1} \eta_t(1 - c_1\eta_t)\mathbb{E}(\|\nabla L(\theta^t)\|^2) \le \mathbb{E}\{L(\theta^0) - L(\theta^{t+1})\} + \sum_{t=0}^{T-1} c_2\eta_t^2.$$

For a sufficiently small $\eta_t$-sequence such that $\eta_t \le 1/(2c_1)$ for all $t$, we have

$$\min_{t=0,\ldots,T-1} \mathbb{E}(\|\nabla L(\theta^t)\|^2) \sum_{t=0}^{T-1} \eta_t/2 \le \sum_{t=0}^{T-1} \frac{\eta_t}{2}\mathbb{E}(\|\nabla L(\theta^t)\|^2) \le \mathbb{E}\{L(\theta^0) - L(\theta^T)\} + \sum_{t=0}^{T-1} c_2\eta_t^2.$$

As a result,

$$\min_{t=0,\ldots,T-1} \mathbb{E}(\|\nabla L(\theta^t)\|^2) \le \frac{2\mathbb{E}\{L(\theta^0) - L^*\}}{\sum_{t=0}^{T-1} \eta_t} + \frac{2c_2\sum_{t=0}^{T-1}\eta_t^2}{\sum_{t=0}^{T-1}\eta_t},$$

which concludes the proof.

**Proof of Corollary 11.** The first two statements regarding the special choices of the step-size sequence $\eta_t$ are directly obtained from Theorem 10. As for the gradient descent, when $\sigma = 0$, the constant $c_2$ in the proof of Theorem 10 disappears. This leads to

$$\min_{t=0,\ldots,T-1} \mathbb{E}(\|\nabla L(\theta^t)\|^2) \le \frac{2\mathbb{E}\{L(W_0) - L^*\}}{\sum_{t=0}^{T-1}\eta_t},$$

which completes the proof.

# D   Experimental Details

For completeness, we state the algorithms of 'FedAvg', 'FairFed', and 'LFT' in Algorithm 2, Algorithm 3, and Algorithm 4, respectively. The algorithm of 'FedLFT' is the same as 'FedAvg' except that the local client performs gradient descent according to the following equation:

$$\theta_k^{t,e} \leftarrow \theta_k^{t,e-1} - \eta\nabla_{\theta_k^{t,e-1}}(L_k + \lambda J(F(f(\cdot;\theta_k^{t,e-1}); \widehat{\mathcal{D}}_k))).$$

---

**Algorithm 2** ('FedAvg') Federated Average

---

**Input:** Communication rounds $T$, learning rate $\eta$, local training epochs $E$.
**System executes:**

   Initialize the global model parameters $\theta^0$
   **for** each communication round $t = 1, 2, \dots T$ **do**
      **for** each client $k = 1, \dots, K$ **in parallel do**
         Receive the model parameters $\theta_k^{t,0} = \theta^{t-1}$ from the server
         $\theta_k^{t,E} \leftarrow \textbf{ClientUpdate}(\theta_k^{t,0}, Z)$
      **end**
      Server update global model $\theta^t \leftarrow \sum_k w_k \theta_k^{t,E}$.
   **end**
   Return the final global model $f(\cdot; \theta^T)$
**ClientUpdate** $(\theta_k^{t,0}, Z)$**:**
   **for** each local epoch $e$ from 1 to $E$ **do**
      Perform gradient descent $\theta_k^{t,e} \leftarrow \theta_k^{t,e-1} - \eta \nabla_{\theta_k^{t,e-1}} L_k$
   **end**
   Return $\theta_k^{t,E}$

---

**Algorithm 3** ('FairFed') Fairness-aware Federated Average

---

**Input:** Communication rounds $T$, learning rate $\eta$, local training epochs $E$, hyper-parameter $\beta$.
**System executes:**

   Initialize the global model parameters $\theta^0$
   **for** each communication round $t = 1, 2, \dots T$ **do**
      **for** each client $k = 1, \dots, K$ **in parallel do**
         Receive the model parameters $\theta_k^{t,0} = \theta^{t-1}$ from the server
         $\theta_k^{t,E}, F_k^t, m_k^t \leftarrow \textbf{ClientUpdate}(\theta_k^{t,0}, Z)$
      **end**
      Calculate global fairness $F^t \leftarrow \sum_k w_k m_k^t$
      Aggregation weights $w_k^t \leftarrow \exp(-\beta |F^t - F_k^t|) w_k$
      Server update global model $\theta^t \leftarrow \sum_k w_k^t \theta_k^{t,E}$.
   **end**
   Return the final global model $f(\cdot; \theta^T)$
**ClientUpdate** $(\theta_k^{t,0}, Z)$**:**
   **for** each local epoch $e$ from 1 to $E$ **do**
      Perform any bias mitigation algorithm to this local client
   **end**
   Calculate the local fairness $F_k^t \leftarrow F(f(\cdot; \theta_k^{t,E}), \widehat{\mathcal{D}}_k)$
   Calculate the global fairness component $m_k^t$ `/* See Ezzeldin et al. (2021)`     `*/`
   Return $\theta_k^{t,E}, F_k^t, m_k^t$

---

---

**Algorithm 4** ('LRW') Locally reweighing

---

**Input:** Communication rounds $T$, learning rate $\eta$, local training epochs $E$, penalty parameter $\lambda$.

**System executes:**

    Initialize the global model parameters $\theta^0$ **for** each communication round $t = 1, 2, \ldots T$ **do**

        **for** each client $k = 1, \ldots, K$ **in parallel do**

            Receive the model parameters $\theta_k^{t,0} = \theta^{t-1}$ from the server

            $\theta_k^{t,E} \leftarrow \textbf{ClientUpdate}(\theta_k^{t,0}, Z)$

        **end**

        Server update global model $\theta^t \leftarrow \sum_k w_k \theta_k^{t,E}$.

    **end**

    Return the final global model $f(\cdot; \theta^T)$

**ClientUpdate** $(\theta_k^{t,0}, Z)$**:**

    **for** each local epoch $e$ from 1 to $E$ **do**

        Assign each data point a score associated with its sensitive attribute `/* See Abay et al. (2020)`
            `*/`

        Perform ordinary gradient descent on the weighted loss function

    **end**

    Return $\theta_k^{t,E}$

---

