# OpenReview forum: "Mitigating Group Bias in Federated Learning: Beyond Local Fairness"
_TMLR — Accepted by TMLR_

### Review · Reviewer_qxFB · 2024-05-23

**Summary Of Contributions:**

The paper proposes a framework to mitigate bias at the level of local clients before aggregation. The goal is to mitigate the group bias. Detailed theoretical analysis is provided to support the foundation of the algorithm. Overall, this paper is good.

**Audience:**

Yes

**Broader Impact Concerns:**

Not applicable.

**Claims And Evidence:**

Yes

**Requested Changes:**

+ Elaborate on Definition 1.
+ Add more references related to group fairness in federated learning.
+ Clearly state the difference between the proposed work and [4].
+ Highlight the best performed models in Table 3 for clarity.

**Strengths And Weaknesses:**

Strengths:

+ Group fairness in Federated Learning (FL) is a very important and often underestimated area. This paper makes significant progress by addressing group fairness in FL.
+ The theoretical results presented in this paper are interesting and sound.

Weaknesses:

- Definition 1 is not very clear. Why is a ratio used?
- The literature review is not thorough. This paper omits some important references or benchmark models, including but not limited to:
[1] Li, T., Hu, S., Beirami, A., & Smith, V. (2021, July). Ditto: Fair and robust federated learning through personalization. In International Conference on Machine Learning (pp. 6357-6368). PMLR.
[2] Yue, X., Nouiehed, M., & Al Kontar, R. (2023). Gifair-fl: A framework for group and individual fairness in federated learning. INFORMS Journal on Data Science, 2(1), 10-23.
[3] Papadaki, A., Martinez, N., Bertran, M., Sapiro, G., & Rodrigues, M. (2022, June). Minimax demographic group fairness in federated learning. In Proceedings of the 2022 ACM Conference on Fairness, Accountability, and Transparency (pp. 142-159).
[4] Zeng, Y., Chen, H., & Lee, K. (2021). Improving fairness via federated learning. arXiv preprint arXiv:2110.15545.

Furthermore, the paper's formulation resembles that of [4]. The authors should elaborate more on the differences. Additionally, it is recommended that the authors include more references related to group fairness in federated learning.

---

> ### Author Response · Authors · 2024-06-10
> **Response 1**
>
> We sincerely thank Reviewer qxFB for their invaluable feedback, which has significantly contributed to enhancing the quality of our work. We have dedicated considerable effort to addressing every comment. Besides the replies below, we have also uploaded a revised manuscript incorporating all discussions, with major changes highlighted in blue color.
>
> Requested changes:
>
> 1. Elaborate on Definition 1.
>
>     **Response:** Thanks for your insightful suggestions on elaborating and further motivating our Definition 1.
>
>     Our Definition 1 aims to capture the concept of group fairness, which pursues equitable treatment of different groups, such as people of different genders or races. Nevertheless, a mathematical formulation for such a concept is lacking, and most existing works (Caton & Haas, 2020; Mehrabi et al., 2021) explain group fairness using plain words or examples such as statistical parity and equal opportunity.
>
>     To formulate group fairness, we are motivated by the observation that many widely used group fairness metrics adopt the form of the **disparity of model performance between different groups**, especially for a binary group variable. Taking statistical disparity for example, it is the difference of the probability being predicted as positive for individuals from different groups. In other words, when $A$ is a binary group variable taking values in 0 and 1, the statistical disparity is measured by $|P(\hat{Y}=1|A=0)- P(\hat{Y}=1|A=1)|$, where $\hat{Y}$ is the prediction by model $f$. We thus propose that the model performance given a specific group $s$ can be typically expressed as a conditional expectation on the event that $A=s$, namely $E(g(\hat{Y})|A=s)$ for any evaluation function $g$. In the case of statistical disparity, $g(\hat{Y})=1_{\hat{Y}=1}$ and $1_{(\cdot)}$ is the indicator function.
>
>     Then, why our Definition 1 takes form of the difference of two ratios? There are two critical reasons. **First,** the Bayes theorem shows that $E(g(\hat{Y})|A=s)$ can always be written as the ratio of two expectations, therefore a group fairness metric can be naturally written as the form in Definition 1. **Second,** the increase in the degree of freedom (the additionally introduced $b$ and $d$ in Def 1) enables a finer grid analysis on the relationship between global and local fairness. While global fairness and local fairness are not aligned for general group-fairness metric, our developed theory proves that they are closely related if $b$ and $d$ are irrelevant to the model $f$.
>
>
>
>     Simon Caton and Christian Haas. Fairness in machine learning: A survey. arXiv preprint arXiv:2010.04053, 2020.
>
>     Ninareh Mehrabi, Fred Morstatter, Nripsuta Saxena, Kristina Lerman, and Aram Galstyan. A survey on bias and fairness in machine learning. ACM Computing Surveys (CSUR), 54(6):1–35, 2021.
>
>
>
> 2. Add more references related to group fairness in federated learning.
>
>    **Response:** Your suggestion regarding the related literature is greatly appreciated. We have cited all mentioned references in the revision.

---

> ### Author Response · Authors · 2024-06-10
> **Response 2**
>
> 3. Clearly state the difference between the proposed work and [4].
>
>    **Response:** We thank the reviewer for your helpful suggestion. We have discussed [4] in the past work section. Below, we further elaborate our discussion and comparison with this work. In short, the scope and contribution of Zeng et al. (2021) is different from ours.
>
>    Theoretically, Zeng et al. (2021) studied the problem of whether federated learning is helpful for decentralized fairness learning. In particular, they show that in terms of fairness-utility trade-off, training locally fair models without communicating with each other and assembling them is worse than LFT (as defined in our paper), and LFT is worse than centralized fair learning. **In contrast**, we analyze the relationship between the fairness of local models and the global model in federated learning, which is untouched by Zeng et al. (2021).
>
>    As for the algorithm, Zeng et al. (2021) proposed FedFB that combines federated learning with FairBatch (a fair training method originally proposed in the centralized setting). FedFB solves a bi-level optimization problem to optimize the fairness criterion, where the outer loop adaptively chooses training batches that fairly represent the training data, and the inner loop optimizes model parameters. **In contrast,** our proposed FedGFT is inspired by our developed theory (Theorem 8), namely the gradient of global fairness can be obtained from local fairness-related quantities. As a result, FedGFT can seamlessly integrate with any conventional federated learning algorithms such as FedAvg and FedProx, by only slightly adapting the local objective. Compared to FedFB, FedGFT offers a simpler optimization procedure without the need of bi-level optimization. Moreover, our methods apply to a broader class of fairness metrics than FedFB. For example, both conditional equal opportunity (Beutel et al., 2019) and balance for positive and negative class (Kleinberg et al., 2017) are proper group-based fairness metrics and therefore can be optimized using FedGFT, while they do not meet the assumptions made by FedFB.
>
>
>
>    [4] Zeng, Y., Chen, H., & Lee, K. (2021). Improving fairness via federated learning. arXiv preprint arXiv:2110.15545.
>
>    Alex Beutel, Jilin Chen, Tulsee Doshi, Hai Qian, Allison Woodruff, Christine Luu, Pierre Kreitmann, Jonathan Bischof, and Ed H Chi. Putting fairness principles into practice: Challenges, metrics, and improvements. In Proceedings of the 2019 AAAI/ACM Conference on AI, Ethics, and Society, pages 453–459, 2019.
>
>    Jon Kleinberg, Sendhil Mullainathan, and Manish Raghavan. Inherent trade-offs in the fair determination of risk scores. Innovations in Theoretical Computer Science, 2017.
>
>
> 4. Highlight the best performed models in Table 3 for clarity.
>
>     **Response:** Thank you for your suggestion on improving the paper readability. We have marked the best model’s score in bold in the revision.

---

### Review · Reviewer_eYUo · 2024-06-02

**Summary Of Contributions:**

This paper presents theoretical foundations for fair federated training. The authors categorize different fairness objectives and define the class of proper group-based fairness metrics that can be optimized for locally. They prove that global model fairness using a proper group-based fairness metric is upper bounded by local fairness and data heterogeneity across clients, which gives support for the success of locally fair training in homogeneous settings. They present an algorithm, FedGFT, for globally fair federated training achieved by augmenting the local client objective function to include a fairness regularization term. The paper includes some empirical results on a few datasets commonly used in the fairness literature.

**Audience:**

Yes

**Claims And Evidence:**

Yes

**Requested Changes:**

Critical changes:
*  While the theoretical sections of the paper are especially strong, the paper lacks empirical analysis that could greatly strengthen the work. The experimental section and discussion could be bolstered by the following changes:
    * State how evaluation is done.
    * State the default hyperparameters used in the results presented in Table 3 in section 6.2.
    * The discussion in 6.3 is significantly lacking, does not reflect what is shown in the figures and does not provide the strongest experimental support for the method presenting. For example, Figure 1 demonstrates that accuracy and bias improve with increasing number of epochs. This is not reflected in the text under 6.3 “Number of epochs.” Furthermore, there is no baseline for comparing how varying the number of epochs impacts performance (i.e. in this setting does more epochs of training also benefit FedAvg?).
    * Pages 9-11 only include figures, without enough discussion or analysis warranting their inclusion in the paper. Consider including just a few plots that are strictly necessary for assessing the performance of FedGFT across settings of interest. Arguably what is shown in Figures 1, 2 and 3 is simply the hyperparameter search, which belongs in the appendix.
    * Far more interesting and informative ablations to depict and discuss in the main text include: how does FedGFT handle extreme forms of client heterogeneity (pure clients from a single group)? How does the data partitioning (to many more clients, to different types of clients, to uneven numbers of examples per client) change the results across all methods of comparison?
    * Expand 6.2 to discuss the characteristics of settings in which FedGFT succeeds.
* Discuss the added computation and communication costs of FedGFT.

Minor changes:
* Bold the best value for each setting in the table so that it’s more readable and clear how the methods compare.
* Specify which of the methods discussed in Section 7 “Past works” are included in empirical comparison. Include the names in the table used to refer to these methods, and bold them in the text.
* Table 6: include results for FedAvg on the pure group.
* Include discussions about what assumptions are made of the federated setting in which FedGFT is applicable and has been applied to in the experiments done.

**Strengths And Weaknesses:**

Strengths:
* Well-written with sound reasoning that is easy to follow. Includes discussion of theoretical proofs with implications of each result. For example, in addition the the proof provided, demonstrates via a simple illustrative example that local fairness does not imply group fairness.
* Proves that locally fair training and heterogeneity upper-bound the resulting global model fairness given by a proper group-based fairness metric.
* Presents an algorithm FedGFT for globally fair training with a proper group-based fairness metric, which uses strictly local summary statistics to minimize the fairness-penalized empirical loss of the global model.
* Strong theoretical support for the globally optimal fair federated training algorithm, FedGFT.
* FedGFT is applicable more generally to any proper group-based fairness metric, beyond those presented in the paper.
* Integrates well with existing federated learning algorithms.
* Does not require bi-level optimization, as is required by several other fair federated algorithms.

Weaknesses:
* For every federated round, FedGFT requires another round of client computation and communication to compute the constant C, which is then used in the fairness-regularized client update. This is 2x the communication steps (albeit much smaller packets being sent).
* Experimental setup details missing:
    * What are the client and server optimizers used for FedAvg and variants?
    * What cohort size is used in each federated round (all clients, or some sampling)?
    * How are evaluations carried out? Is evaluation done on just the test portion of each dataset, or is each test dataset partitioned with $\alpha$ so that federated evaluation can be performed?
* Lacking empirical analysis:
    * Very little discussion of the experimental results.
    * Ablations presented in the main text are simply a hyperparameter search and do not provide support for the method presented.
* While FedGFT is applicable to the cross-silo federated learning setting, it is unclear how FedGFT might perform in the cross-device federated learning setting, where there is likely to be partial client participation and it may not be possible to sample the same set of clients for computing C as for computing the client updates.
* How might FedGFT extend beyond binary classification with a single sensitive attribute?
* How should $\lambda$ be chosen? Is there any relation to the underlying data heterogeneity $\alpha$?

---

> ### Author Response · Authors · 2024-06-10
> **Response 1**
>
> We sincerely thank Reviewer eYUo for their invaluable feedback, which has significantly contributed to enhancing the quality of our work. We have dedicated considerable effort to addressing every comment. Besides the replies below, we have also uploaded a revised manuscript incorporating all discussions, with major changes highlighted in blue color. Thank you again for your time and effort in reviewing our work.
>
> Requested changes:
>
> ## Critical changes:
>
> 1. State how evaluation is done.
>
>    **Response:** We greatly appreciate the insightful suggestions from the reviewer on improving the experiments and the overall quality of our manuscript.
>
>    We evaluate the performance of the final global model on the test dataset. Specifically, the original dataset is first randomly split into three parts, training, validation, and test dataset. The training dataset is further split into disjoint subsets, serving as local datasets of clients. After training is done, the accuracy and fairness metric of the global model is calculated on the test dataset. In this way, the test dataset has the same distribution as the population, namely the weighted average of local clients. Our evaluation process thus properly measures the performance of global model.
>
> 2. State the default hyperparameters used in the results presented in Table 3 in section 6.2.
>
>    **Response:** We have summarized the default hyperparameters in Table 4 in Appendix D.2. In the revision, we have moved this table to Section 6.2. We also cite it in this reply for your reference (see https://openreview.net/forum?id=ANXoddnzct&noteId=cAf1mi0y5j). Note that all clients are involved in each federated round, and we will discuss how to allow the involvement of partial clients in the reply to Point 4 of minor changes.
>
> 3. The discussion in 6.3 is significantly lacking, does not reflect what is shown in the figures and does not provide the strongest experimental support for the method presenting. For example, Figure 1 demonstrates that accuracy and bias improve with increasing number of epochs. This is not reflected in the text under 6.3 “Number of epochs.” Furthermore, there is no baseline for comparing how varying the number of epochs impacts performance (i.e. in this setting does more epochs of training also benefit FedAvg?).
>
> 4. Pages 9-11 only include figures, without enough discussion or analysis warranting their inclusion in the paper. Consider including just a few plots that are strictly necessary for assessing the performance of FedGFT across settings of interest. Arguably what is shown in Figures 1, 2 and 3 is simply the hyperparameter search, which belongs in the appendix.
>
> 5. Far more interesting and informative ablations to depict and discuss in the main text include: how does FedGFT handle extreme forms of client heterogeneity (pure clients from a single group)? How does the data partitioning (to many more clients, to different types of clients, to uneven numbers of examples per client) change the results across all methods of comparison?
>
>    **Response to 3, 4, and 5:** Following your comments, we plan to move the experiments of hyper-parameter to appendix, and move the ablations you have mentioned in point 5 to the experiment section.
>
>    **Discussion on the choice of hyper-parameters for FedGFT.**  We rerun the experiments that vary each hyper-parameter with communication round $T=50$​, which ensures that the model achieves convergence at the end of training. On the COMPAS dataset, we observe that the learning rate and number of epochs will jointly determine the performance of FedGFT. When the learning rate and the number of epochs are increased, it will first accelerate model convergence (less communication round), since there are more local updating steps in each round and the step size is increased. However, when the learning rate or the number of epochs is too large, it will impede model training instead. In particular, a large learning rate will cause the local model to oscillate around the minimizer and hence hard to converge, which is a known issue of the classical neural network training using an optimizer with a fixed learning rate. On the other hand, a large number of epochs can amplify the difference among local models at each communication round, causing slow convergence for the global model. Nevertheless, according to the above analysis, we may accommodate the choice of hyper-parameter by using an optimizer with a shrinking learning rate and an objective that can address data heterogeneity. Lastly, it is recommended to choose those hyper-parameters through cross-validation.
>
>     (Continued at the reply to this comment, see https://openreview.net/forum?id=ANXoddnzct&noteId=wwuk65p5Lr)

---

> > ### Author Response · Authors · 2024-06-16
> > **Default hyperparameters used in the results presented in Table 3 in section 6.2.**
> >
> > Table 4: Default training hyperparameters.
> >
> >    |      Dataset      | Adult  | COMPAS |   CelebA    |
> >    | :---------------: | :----: | :----: | :---------: |
> >    |   Architecture    | Linear | Linear |  ResNet18   |
> >    | Number of clients |   10   |   10   |     10      |
> >    |    Comm. round    |   20   |   20   |     20      |
> >    |    Batch size     |  256   |  256   |     64      |
> >    |       Epoch       |   1    |   1    |      1      |
> >    |     Optimizer     |  ADAM  |  ADAM  |    ADAM     |
> >    |   Learning rate   | 0.002  | 0.002  |    0.001    |
> >    |     Scheduler     |  N/A   |  N/A   | MultistepLR |
> >    |   Weight decay    |  N/A   |  N/A   |     0.1     |

---

> ### Author Response · Authors · 2024-06-10
> **Response 2**
>
> 6. Expand 6.2 to discuss the characteristics of settings in which FedGFT succeeds.
>
>    **Response:** From Table 3, we find that the proposed FedGFT can achieve a small fairness bias with comparable accuracy to FedAvg across three datasets and different degrees of data heterogeneity. This is aligned with our developed theory, as FedGFT is designed to directly optimize the global objective, thus the data heterogeneity has little influence on its performance.
>
> 7. Discuss the added computation and communication costs of FedGFT.
>
>    **Response:** Analysis of additional computation and communication costs of FedGFT. Compared to classical federated learning algorithms, FedGFT involves an additional step of updating a fairness-related constant at the beginning of each federated round, as shown in Algorithm 1. In particular, this constant update step calculates $F_k=a_k/b-c_k/d$ for each client $k$, where $a_k$ and $c_k$ are specified by the fairness criterion and the local model, while $b$ and $d$ are two universal constants. Moreover, $a_k$ and $c_k$ can be obtained via forward propagating the local model on the local dataset for once. As a result, the additional computation cost is one round of forward propagation step, while the computation cost for a typical federated learning algorithm at each federated round is $E$ rounds of forward and backward propagation steps, where $E$​ is the number of epochs.
>
>    As for the communication cost, our originally proposed Algorithm 1 has an addition round to synchronize the constant, as pointed out by the reviewer. Nevertheless, this communication round only transmits a single scaler, therefore the cost is much smaller than transmitting model parameters. Furthermore, a possible way to eliminate this additional communication round is synchronizing the constant at the same round as model parameters. Specifically, the local clients can calculate an approximation of $F_k$ at the end of their local training round using the local model, and pass it to the global model. In this way, we do not need a separate round of constant update, at the cost of inaccurate calculation of $F_k$.
>
>
> ## Minor Changes
>
> 1. Bold the best value for each setting in the table so that it’s more readable and clear how the methods compare.
>
>    **Response:** Thank you for your suggestion on improving the paper readability. We have marked the best score in bold in the revision.
>
> 2. Specify which of the methods discussed in Section 7 “Past works” are included in empirical comparison. Include the names in the table used to refer to these methods, and bold them in the text.
>
>    **Response:** We have incorporated your suggestion in the revision.
>
> 3. Table 6: include results for FedAvg on the pure group.
>
>    **Response:** Following your suggestions in the critical changes, we have now moved the pure-client case to the experiment section. We have completed the experiments for FedAvg on the pure group, and the results are reported in the table in this reply (see https://openreview.net/forum?id=ANXoddnzct&noteId=wwuk65p5Lr).
>
> 4. Include discussions about what assumptions are made of the federated setting in which FedGFT is applicable and has been applied to in the experiments done.
>
>    **Response:** In our originally proposed FedGFT and experiments, there are two assumptions: (1) the fairness metric is proper, (2) all clients are involved in each federated round. Here, a proper group-based fairness metric plays an essential role for FedGFT, while the second condition can be relaxed. Specifically, for cross-device federated learning, we can apply FedGFT as long as clients are independently sampled in each round. Recall that our ultimate goal is optimizing the penalized objective Eq. (2), and FedGFT shows that its gradient can be written as the linear combination of gradients from local clients. When clients are sampled independently, then we can replace the whole gradient with its stochastic counterpart, namely the combination of gradients from those independently sampled clients. This is akin to the relationship between the gradient descent algorithm and the stochastic gradient descent algorithm.

---

> ### Author Response · Authors · 2024-06-10
> **Response 3**
>
> ## Weakness
>
> 1. How might FedGFT extend beyond binary classification with a single sensitive attribute?
>
>    **Response:** For a multi-class group variable, say, $A \in \{1,\dots,C\}$ for some positive integer $C\geq 2$, one possible direction to generalize the definition of group-based fairness metrics is considering
>    $$
>    F(f, \mathcal{D}) = \sum_{1\leq i< j\leq C}\biggl| \frac{a_i(f, \mathcal{D})}{b_i(f, \mathcal{D})}- \frac{a_j(f, \mathcal{D})}{b_j(f, \mathcal{D})} \biggr|,
>    $$
>      where $a_i(f, \mathcal{D})$ and $b_i(f, \mathcal{D})$ are some expectations on the event $\{A=i\}$. Accordingly, a proper group-based fairness metric is one such that $b_i$'s are degenerated with respect to $f$. We believe that by using the same techniques as the current work, we can derive similar results as Theorem 2, 4, and 7.
>
>    If there is more than one sensitive attribute, say, $A_s \in \{1,\dots,C_s\}$ for $s=1,\dots,S$, we can analogously consider
>
>    $$
>    F(f, \mathcal{D}) = \sum_{1\leq s\leq S}\sum_{1\leq i< j\leq C_S}\biggl| \frac{a_i^s(f, \mathcal{D})}{b_i^s(f, \mathcal{D})}- \frac{a_j^s(f, \mathcal{D})}{b_j^s(f, \mathcal{D})} \biggr|.
>    $$
>    There are also other ways to formulate the fairness metric for multi-class attributes, such as the variance of ratios $a_i/b_i$, instead of pairwise differences. Nevertheless, the proposed FedGFT can be extended and applied to those scenarios as long as $a/b$ can be obtained using local statistics.
>
> 2. How should $\lambda$ be chosen? Is there any relation to the underlying data heterogeneity $\alpha$​​?
>
>    **Response:** In practice, a common way for choosing hyper-parameter is through cross-validation. Typically, a higher $\lambda$ leads to a model with smaller fairness bias but worse accuracy. The choice of $\lambda$ is unrelated to $\alpha$ since FedGFT optimizes the global objective directly.

---

> ### Author Response · Authors · 2024-06-16
> **Response to 3, 4, and 5, Continued (1/2)**
>
> **Response to 3, 4, and 5, Continued.**
>
> **Experiments with different kinds of data and client heterogeneity.**
>
> We first conduct experiments on the extreme forms of client heterogeneity (pure clients from a single group). The comparison between FedAvg and FedGFT is summarized in Table 5.
>
> Table 5: The average accuracy and bias (standard error in parentheses) on the COMPAS dataset under two fairness metrics. Pure group represents the situation where clients are purely from one group; mixed group represents the case where clients have data from both groups; and $\lambda$ is the penalty parameter.
>
> | Method                       | Acc        | SP         | EOP        |
>    | ---------------------------- | ---------- | ---------- | ---------- |
>    | FedAvg (Mixed group)                | 66.7 (0.2) | 12.4 (0.5) | 12.0 (0.5) |
>    | FedAvg (Pure Group)          | 66.6 (0.2) | 6.9 (0.5)  | 6.2 (0.4)  |
>    | FedGFT (Mixed group, $\lambda=20$)  | 66.1 (0.2) | 0.5 (0.0)  | 0.7 (0.0)  |
>    | FedGFT (Pure group, $\lambda=10$)   | 65.5 (0.3) | 2.4 (0.3)  | 2.2 (0.2)  |
>    | FedGFT (Pure group, $\lambda=20$ )  | 64.4 (0.4) | 1.7 (0.2)  | 1.4 (0.1)  |
>    | FedGFT (Pure group, $\lambda=100$ ) | 59.0 (0.8) | 1.0 (0.1)  | 0.5 (0.1)  |
>
> We are also running experiments comparing the performance of all five methods under different numbers of clients and numbers of examples per client. We will update the results as soon as the experiments are completed.
>
> **Experiments with different client heterogeneity levels**
> We then investigate the learning algorithm's performance given the following two scenarios: 1. there are many more clients. 2. The number of data points owned by each client is highly unequal. We conduct experiments on the COMPAS dataset and compare all six methods except FedFB.
>
> For the first scenario, with other settings remaining the same, the number of clients is 100. That is, each client has tens of data points on average. The results are reported in Table 6. Compared to the case where only 10 clients are involved, FedGFT is still the algorithm with the smallest bias. Moreover, the accuracy drop by FedGFT is similar or smaller to other fairness-aware algorithms.
>
> Table 6: The average accuracy and bias (standard error in parentheses) on COMPAS datasets when there are 100 clients.
> | $\alpha$ | Method  | Acc ($\uparrow$) | SP ($\downarrow$) | EOP ($\downarrow$) |
> |----------|---------|------------------|-------------------|--------------------|
> | 0.5      | FedAvg  | 66.0 (0.2)       | 9.0 (0.8)         | 9.3 (2.0)          |
> |          | LRW     | **66.9 (0.4)**     | 9.6 (0.5)         | 6.5 (1.3)          |
> |          | FairFed | 61.9 (1.7)       | 5.3 (0.5)         | 3.9 (0.8)          |
> |          | FedLFT  | 61.6 (0.9)       | 0.9 (0.1)         | 1.7 (0.3)          |
> |          | FedGFT  | 62.2 (1.6)       | **0.8 (0.1)**        | **1.1 (0.2)**       |
> | 5.0      | FedAvg  | **67.4 (0.3)**      | 12.9 (0.2)        | 13.0 (0.2)         |
> |          | LRW     | 66.3 (0.3)       | 3.7 (0.2)         | 3.5 (0.2)          |
> |          | FairFed | 66.7 (0.3)       | 3.6 (0.2)         | 3.0 (0.2)          |
> |          | FedLFT  | 66.2 (0.4)       | 1.3 (0.1)         | 1.0 (0.1)          |
> |          | FedGFT  | 66.0 (0.3)       | **0.5 (0.0)**         | **0.7 (0.0)**        |
> | 100.0    | FedAvg  | **67.6 (0.4)**      | 14.2 (0.1)        | 14.1 (0.1)         |
> |          | LRW     | 66.9 (0.2)       | 2.5 (0.1)         | 2.6 (0.1)          |
> |          | FairFed | 66.4 (0.3)       | 2.3 (0.1)         | 2.3 (0.2)          |
> |          | FedLFT  | 66.5 (0.2)       | 1.1 (0.1)         | 1.1 (0.1)          |
> |          | FedGFT  | 66.8 (0.2)       | **0.5 (0.0)**       | **0.5 (0.0)**      |
>
>
> (Continued in https://openreview.net/forum?id=ANXoddnzct&noteId=6ndKGLjLks)

---

> > ### Author Response · Authors · 2024-06-17
> > **Response to 3, 4, and 5, Continued (2/2)**
> >
> > Next, we consider the scenario where each client can have different number of data points. We modify the data split step as follows. First, we generate the proportion of data points for each client from a Dirichlet distribution $Dir(\alpha)$. Then, for each combination of sensitive attribute $A$ and response $Y$, we randomly assign the corresponding proportion of data points to each client. It means that the data is completely homogeneous to each client, which is the most favorable scenario for locally fair training methods. The parameter $\alpha$ takes values in $0.5, 5,$ and $100$, and a higher $\alpha$ indicates more evenly distributed numbers of data points. For example, the median ratio of the largest and smallest number of data points is around 400 when $\alpha = 0.5$, while this ratio is 1.3 for $\alpha=100$.
> >
> > The results of clients with varying numbers of data points are reported in Table 7. Clearly, FedGFT has the smallest bias even in the homogeneous setting, with comparable accuracy to FedAvg. The results, together with the experiments under different data heterogeneity levels, demonstrate that FedGFT is a robust and effective fairness-aware federated learning algorithm.
> >
> > Table 7: The average accuracy and bias (standard error in parentheses) on COMPAS datasets when each client's number of data points can vary.
> >
> > | $\alpha$ | Method  | Acc ($\uparrow$) | SP ($\downarrow$) | EOP ($\downarrow$) |
> > |----------|---------|------------------|-------------------|--------------------|
> > | 0.5      | FedAvg  | 65.7 (0.2)       | 22.4 (0.9)        | 23.3 (0.8)         |
> > |          | LRW     | **66.4 (0.3)**      | 3.6 (0.7)         | 2.6 (0.3)          |
> > |          | FairFed | 65.8 (0.3)       | 2.2 (0.6)         | 1.8 (0.4)          |
> > |          | FedLFT  | 64.8 (0.4)       | 1.4 (0.4)         | 3.8 (0.8)          |
> > |          | FedGFT  | 65.4 (0.3)       | **0.3 (0.0)**       | **0.4 (0.1)**       |
> > | 5.0      | FedAvg  | 66.3 (0.3)       | 27.1 (0.2)        | 26.4 (0.4)         |
> > |          | LRW     | **67.0 (0.2)**     | 5.4 (0.4)         | 3.1 (0.3)          |
> > |          | FairFed | 65.9 (0.4)       | 4.3 (0.8)         | 0.9 (0.1)          |
> > |          | FedLFT  | 65.3 (0.3)       | 0.9 (0.1)         | 1.4 (0.3)          |
> > |          | FedGFT  | 65.3 (0.2)       | **0.3 (0.0)**      | **0.4 (0.0)**       |
> > | 100.0    | FedAvg  | 65.5 (0.3)       | 26.0 (0.1)        | 26.5 (0.1)         |
> > |          | LRW     | **66.4 (0.3)**     | 3.4 (0.1)         | 3.3 (0.2)          |
> > |          | FairFed | 65.6 (0.3)       | 2.7 (0.4)         | 2.5 (0.3)          |
> > |          | FedLFT  | 65.9 (0.3)       | 1.0 (0.2)         | 0.7 (0.1)          |
> > |          | FedGFT  | 65.4 (0.2)       | **0.3 (0.0)**       | **0.3 (0.0)**       |

---

### Review · Reviewer_wLz4 · 2024-06-04

**Summary Of Contributions:**

This paper explores global group fairness in federated learning, focusing on statistical parity and equal opportunity metrics. It proposes a regularised learning objective for achieving global fairness, a FedAvg-inspired algorithm to solve the objective and its convergence guarantees for a single local epoch. It also analyses the differences between local and global model fairness notions in FL settings.

**Audience:**

Yes

**Claims And Evidence:**

No

**Requested Changes:**

Please see the weaknesses section.

**Strengths And Weaknesses:**

**Strengths:**
* Studying and formalizing how local and global fairness notions connect/depart is an important area in federated learning literature which has attracted much interest.
* The proposed objective and solution are simple and can be easily integrated into other approaches (please see some concerns about this below).

**Weaknesses:**
The experimental results seem to require substantial improvements and some claims throughout the paper need more clarification and/or adjustments to be valid. Also, discussion and comparison to some relevant works are missing.

In particular:

*(**Critical**) Fairness and validity of empirical comparisons/results:*

 * Figures $1$ and $2$ use a learning rate equal to $0.002$ but *Figure $3$ proves that this learning rate is not good enough* compared to other options. This option seems to be used for all baseline methods in Table 1 as well, which raises the questions of why this was selected and whether this option is good for training other methods.

  * The communication rounds are set to $20$ for each method across all datasets. Again, it is not clear whether this is sufficient for each method to converge given that also the local epochs for all approaches are set to $1$ and the learning rate is small.

 * In Figure $1$, for a fixed number of rounds, the largest number of local epochs seem to perform best in most cases (especially wrt average accuracy), even for high heterogeneity which is unusual.

*Clarifications on claims:*
 * "Note that each epoch will divide local datasets into several batches and thus perform multiple steps of local update." Is this the correct setting for all approaches?

* FedGFT adapts the local objective which indeed can be incorporated into various existing algorithms. However, this doesn't necessarily mean that fairness guarantees will automatically apply since it depends on the optimization objective of the algorithm it's added to.

* "To speed up the training process, we randomly select $10,000$ images for training and $6,000$ for testing in each replicate." Does this mean that the proposed approach scales badly with the data dimensionality?

* The utility of applying FedGFT in situations where clients comprise a single group is unclear. (Table $6$): since FedGFT aims for global group fairness shouldn't the results for FedGFT with mixed groups align with at least one of the pure group versions?

*Relevant related works:*
 * The authors should discuss how their findings compare to [1], which also formally explores the connections between global and local fairness in FL.

 * How does FedGFT empirically compare to [2] that can recover demographic parity or equal opportunity fairness notions?

 [1]  Faisal Hamman, Sanghamitra Dutta, Demystifying Local \& Global Fairness Trade-offs in Federated Learning Using Partial Information Decomposition.

 [2] Shengyuan Hu, Zhiwei Steven Wu, Virginia Smith, Fair Federated Learning via Bounded Group Loss.

---

> ### Author Response · Authors · 2024-06-11
> **Response 1**
>
> We sincerely thank Reviewer wLz4 for their invaluable feedback, which has significantly contributed to enhancing the quality of our work. We have dedicated considerable effort to addressing every comment. Besides the replies below, we have also uploaded a revised manuscript incorporating all discussions, with major changes highlighted in blue color. Thank you again for your time and effort in reviewing our work.
>
> # Weaknesses
>
> ## Fairness and validity of empirical comparisons/results:
>
> 1. Figures 1 and 2 use a learning rate equal to 0.002 but Figure 3 proves that this learning rate is not good enough compared to other options. This option seems to be used for all baseline methods in Table 1 as well, which raises the questions of why this was selected and whether this option is good for training other methods.
>
> 2. The communication rounds are set to 20 for each method across all datasets. Again, it is not clear whether this is sufficient for each method to converge given that also the local epochs for all approaches are set to 1 and the learning rate is small.
>
> 3. In Figure 1, for a fixed number of rounds, the largest number of local epochs seem to perform best in most cases (especially wrt average accuracy), even for high heterogeneity which is unusual.
>
> **Response to 1, 2, and 3:** We greatly appreciate the insightful comments from the reviewer on our experiments, which help improve the overall quality of our manuscript.
>
> First, to address your concern on whether the default hyper-parameters are properly chosen so that each method achieves convergence, we rerun the experiments with an increased learning rate, number of epochs, and communication rounds. In particular, we use $\eta=0.01$, $E=3$ and $T=50$​​. The accuracy v.s. rounds curve in the training phase of a pilot experiment implies that all methods have reached convergence under this combination. We report the results on the COMPAS and Adult datasets as a table in the reply to this comment (see, https://openreview.net/forum?id=ANXoddnzct&noteId=W71UiJzLbW). From the table, FedGFT indeed has the best performance regarding decreasing the bias among all methods with sufficient training.
>
> Second, we provide the following discussion on the choice of hyper-parameters for FedGFT, after rerunning the ablation experiments that vary each hyper-parameter with communication round $T=50$, which ensures that the model achieves convergence at the end of training. On the COMPAS and Adult datasets, we observe that the learning rate and number of epochs will jointly determine the performance of FedGFT. When the learning rate and the number of epochs are increased, it will first accelerate model convergence (less communication round), since there are more local updating steps in each round and the step size is increased. However, when the learning rate or the number of epochs is too large, it will impede model training instead. In particular, a large learning rate will cause the local model to oscillate around the minimizer and hence hard to converge, which is a known issue of the classical neural network training using an optimizer with a fixed learning rate. On the other hand, a large number of epochs can amplify the difference among local models at each communication round, causing slow convergence for the global model. Nevertheless, according to the above analysis, we may accommodate the choice of hyper-parameter by using an optimizer with a shrinking learning rate and an objective that can address data heterogeneity. Lastly, it is recommended to choose those hyper-parameters through cross-validation.
>
> The above discussion also addresses your question about the number of local epochs. The original experiments in Figure 1 are done with a small learning rate, thus an increased number of epochs can improve the performance of FedGFT. However, when we increase the learning rate, a large number of local epochs may negatively impact the accuracy and fairness of the trained model by FedGFT.

---

> ### Author Response · Authors · 2024-06-11
> **Response 2**
>
> ## Clarifications on claims:
>
> 1. "Note that each epoch will divide local datasets into several batches and thus perform multiple steps of local update." Is this the correct setting for all approaches?
>
>    **Response:** All methods used in experiments divide local datasets into batches when performing local updates as long as the batch size is no larger than the training sample size, which is the standard way for neural network training.
>
> 2. FedGFT adapts the local objective which indeed can be incorporated into various existing algorithms. However, this doesn't necessarily mean that fairness guarantees will automatically apply since it depends on the optimization objective of the algorithm it's added to.
>
>    **Response:** The idea of plugging in FedGFT to classical federated learning algorithm is that FedGFT adds an additional penalty term for bias. Therefore, it creates a trade-off between the original federated learning algorithm’s objective and fairness, controlled by a hyper-parameter $\lambda$. Specifically, a higher $\lambda$ leads to a model with smaller fairness bias and possibly worse accuracy. Therefore, the user can achieve certain fairness requirements by choosing $\lambda$ properly.
>
> 3. "To speed up the training process, we randomly select 10,000 images for training and 6,000 for testing in each replicate." Does this mean that the proposed approach scales badly with the data dimensionality?
>
>    **Response:** We subsample the CelebA dataset since we only have limited GPU resources, therefore running experiments will spend less time for all methods. This time reduction comes from the shrink of training sample size solely and is unrelated to data dimensionality. The proposed approach empirically spends a similar time compared to FedAvg on all datasets and settings.
>
> 4. The utility of applying FedGFT in situations where clients comprise a single group is unclear. (Table 6): since FedGFT aims for global group fairness shouldn't the results for FedGFT with mixed groups align with at least one of the pure group versions?
>
>    **Response:** We assume that the reviewer is asking why FedGFT with mixed groups performs better than with pure groups. This is due to the extremely high data heterogeneity for the pure group setting. Note that the FedGFT used in experiments is built upon FedAvg, which is not particularly designed for training with heterogeneous clients. Specifically, high data heterogeneity will cause the local clients update their local models in drastically different directions, resulting in a more difficult training process. Therefore, FedGFT has a better performance when clients have mixed groups than pure groups. Please kindly let us know if there is any misunderstanding.

---

> ### Author Response · Authors · 2024-06-11
> **Response 3**
>
> ## Relevant related works:
>
> 1. The authors should discuss how their findings compare to [1], which also formally explores the connections between global and local fairness in FL.
>
>    [1] Faisal Hamman, Sanghamitra Dutta, Demystifying Local & Global Fairness Trade-offs in Federated Learning Using Partial Information Decomposition.
>
>    **Response:** We thank the review for pointing out this related work. In short, the scope and contribution of [1] is different from ours.
>
>    [1] explores the connections between global and local fairness in FL **for statistical disparity** from the perspective of information theory. Specifically, motivated from the definition of statistical disparity, [1] defines global disparity as the mutual information between the model response and the true label, and defines local disparity as the conditional mutual information between the model response and the true label given the sensitive attribute. Benefited from the specific form of those two definitions, authors of [1] derive the sufficient and necessary condition that local disparity implies global disparity, assuming the underlying data distribution is known.
>
>    In contrast, our work focuses on a broad class of fairness metrics, namely the group-based fairness metric, including statistical disparity, equal opportunities, calibration, and more. We show that global fairness in general does not imply local fairness, and vice versa. Moreover, when fairness metric is proper, we provide an upper bound of global fairness by local fairness and data heterogeneity.
>
>    In summary, [1] provides an information–theoretic perspective for studying statistical disparity, while we consider a general fairness criterion and relate global and local fairness through data heterogeneity.
>
>
>
> 2. How does FedGFT empirically compare to [2] that can recover demographic parity or equal opportunity fairness notions?
>
>    [2] Shengyuan Hu, Zhiwei Steven Wu, Virginia Smith, Fair Federated Learning via Bounded Group Loss.
>
>    **Response:** We are happy to compared with [2] mentioned by the reviewer. Nevertheless, we do not find public code released by the authors of [2]. We also note that [2] is based on the bounded group loss, which is a privacy notion different from ours and can be viewed as a relaxation of demographic parity (DP) or equalized odds (EO).
>
>    To address your concern, we compared our results to the Figure 3 in [2], which reports the values of DP and EO on the COMPAS dataset. In particular, the lowest DP and EO level achieved by [2] is 0.02 while the accuracy drop compared to FedAVG is 0.01. In contrast, our FedGFT can achieve DP and EO level at 0.005, and the accuracy drop compared to FedAVG is within 0.01. Therefore, FedGFT achieves a better fairness-accuracy trade-off than [2] on the COMPAS dataset.

---

> ### Author Response · Authors · 2024-06-14
> **Updated table of experimental results**
>
> Table: Average accuracy and bias (unit is percentage) for five methods.
>
> |       | Dataset |                | COMPAS        |               |                | Adult         |               |
> |-------|---------|----------------|---------------|---------------|----------------|---------------|---------------|
> |       | Metric  | ACC            | SP            | EOP           | ACC            | SP            | EOP           |
> | Alpha | Method  |                |               |               |                |               |               |
> | 0.5   | FedAvg  | **64.2 (0.5)** | 12.8 (1.8)    | 15.2 (2.0)    | 81.8 (0.4)     | 5.7 (1.0)     | 18.3 (3.3)    |
> |       | LRW     | 60.9 (1.3)     | 5.1 (0.8)     | 6.4 (1.4)     | 81.1 (0.5)     | 1.9 (0.4)     | 11.8 (0.5)    |
> |       | FairFed | 61.2 (1.0)     | 5.2 (0.7)     | 3.6 (0.6)     | 80.8 (0.4)     | 2.0 (0.4)     | 10.8 (0.5)    |
> |       | FedLFT  | 61.6 (0.9)     | 2.3 (0.2)     | 4.0 (0.4)     | 79.3 (0.3)     | 3.4 (0.7)     | 2.1 (0.4)     |
> |       | FedGFT  | 63.6 (0.6)     | **0.9 (0.1)** | **1.3 (0.2)** | **81.8 (0.4)** | **0.8 (0.1)** | **1.3 (0.3)** |
> | 5.0   | FedAvg  | **66.7 (0.2)** | 12.4 (0.5)    | 12.0 (0.5)    | **83.3 (0.1)** | 6.2 (0.2)     | 3.6 (0.4)     |
> |       | LRW     | 66.3 (0.3)     | 3.6 (0.2)     | 3.8 (0.3)     | 83.4 (0.1)     | 2.2 (0.1)     | 10.2 (0.1)    |
> |       | FairFed | 65.9 (0.3)     | 3.8 (0.4)     | 3.6 (0.4)     | 83.4 (0.1)     | 2.0 (0.1)     | 10.3 (0.1)    |
> |       | FedLFT  | 65.3 (0.3)     | 1.5 (0.1)     | 2.1 (0.2)     | 82.4 (0.2)     | 2.4 (0.2)     | 2.6 (0.2)     |
> |       | FedGFT  | 66.1 (0.2)     | **0.5 (0.0)** | **0.7 (0.0)** | 83.1 (0.1)     | **0.5 (0.0)** | **0.5 (0.1)** |
> | 100.0 | FedAvg  | **67.1 (0.2)** | 13.4 (0.2)    | 13.5 (0.2)    | 83.3 (0.1)     | 6.9 (0.1)     | 2.3 (0.1)     |
> |       | LRW     | 65.9 (0.2)     | 2.9 (0.1)     | 2.8 (0.1)     | 83.4 (0.1)     | 2.4 (0.0)     | 10.0 (0.1)    |
> |       | FairFed | 65.9 (0.2)     | 2.9 (0.2)     | 2.7 (0.2)     | 83.1 (0.1)     | 2.5 (0.1)     | 9.8 (0.1)     |
> |       | FedLFT  | 65.8 (0.2)     | 1.4 (0.1)     | 2.1 (0.2)     | 83.2 (0.1)     | 2.0 (0.1)     | 2.9 (0.1)     |
> |       | FedGFT  | 66.3 (0.2)     | **0.5 (0.0)** | **0.5 (0.0)** | **83.3 (0.1)** | **0.5 (0.0)** | **0.6 (0.1)** |

---

### Review · Reviewer_sdLf · 2024-06-05

**Summary Of Contributions:**

In this work, the authors present theoretical analysis on the difference between local and global fairness. In addition to that, the authors propose a new fair FL method to optimize the global fairness constraints. Empirical evaluations on three common group fairness benchmarks seem to suggest that the proposed method is effective.

**Audience:**

Yes

**Claims And Evidence:**

No

**Requested Changes:**

See weaknesses above, specifically:
- Explanation of proof of Theorem 2.
- More experiments and comparison with prior works.
- Discussion with Zeng et al.

**Strengths And Weaknesses:**

Strengths:
- The relation between local and global fairness in the context of federated learning is interesting and important.
- The proposed method is simple and seems effective.

Weaknesses:
- Proof of theorem 2 is not straight forward to me. Specifically, in equation 4, it is not obvious to me why the summation could be taken out. Basically, why is it true that $a(f,\sum_k w_kD_k)/b(f,\sum_k w_kD_k)=\sum_kw_ka(f,D_k)/\sum_kw_kb(f,D_k)$. If the authors could provide detailed derivation that will be very helpful.
- On the other hand, consider the case when $K=1$, then $D_k=D$. Theorem 2 seems to fail under this scenario. Am I missing anything?
- Convergence analysis seems to apply only in the setting where $E=1$.
- As the authors mentioned in Remark 9, the authors are using the loss as the surrogate for parity. That means what is actually optimized is not the fairness metric mentioned earlier (e.g. DP, EO). As a result, the convergence bound does not directly transfer to fairness guarantee.  Hence, it's unclear how well the proposed optimization does in terms of provable fairness guarantee.
- Insufficient comparison with prior works. The authors should add comparison with a few prior works[1,2,3] that empirically evaluate on DP and EO in FL settings.
- Zeng et al. present a similar result between local vs global fairness. Could the authors comment on that?

[1] S. Cui, W. Pan, J. Liang, C. Zhang, and F. Wang, “Addressing algorithmic disparity and performance inconsistency in federated learning,” Advances in Neural Information Processing Systems, vol. 34, 2021

[2] Hu, S., Wu, Z. S., & Smith, V. (2024, April). Fair federated learning via bounded group loss. In 2024 IEEE Conference on Secure and Trustworthy Machine Learning (SaTML) (pp. 140-160). IEEE.

[3] Chu, L., Wang, L., Dong, Y., Pei, J., Zhou, Z., & Zhang, Y. (2021). Fedfair: Training fair models in cross-silo federated learning. arXiv preprint arXiv:2109.05662.

---

> ### Author Response · Authors · 2024-06-11
> **Response 1**
>
> We sincerely thank Reviewer sdLf for their invaluable feedback, which has significantly contributed to enhancing the quality of our work. We have dedicated considerable effort to addressing every comment. Besides the replies below, we have also uploaded a revised manuscript incorporating all discussions, with major changes highlighted in blue color. Thank you again for your time and effort in reviewing our work.
>
> ## Weaknesses
>
> 1. Proof of theorem 2 is not straight forward to me. Specifically, in equation 4, it is not obvious to me why the summation could be taken out. Basically, why is it true that $a\left(f, \sum_k w_k D_k\right) / b\left(f, \sum_k w_k D_k\right)=\sum_k w_k a\left(f, D_k\right) / \sum_k w_k b\left(f, D_k\right)$​​. If the authors could provide detailed derivation that will be very helpful.
>
>    **Response:** The summation in $a\left(f, \sum_k w_k D_k\right)$ can be taken out because $a$ is an expectation by Definition 1. Specifically, $a(f,D)$ can be written as $\int g(f,X,Y) D(dXdY)$ for some function $g$. For example, for statistical disparity, $a(f,D) = \int 1_{\\{f(X)>0.5, A=0\\}} D(dXdY)$, where $1_{\\{\cdot\\}}$ is an indicator function. Now, the basic property of integral gives that
>
>    $$
>    a(f, \\sum_k w_k D_k) = \\int g(f,X,Y) \\sum_k w_k D_k(dXdY)
>    \\\\=\\sum_k w_k  \\int g(f,X,Y) D_k(dXdY)
>    \\\\=\\sum_k w_k a(f, D_k).
>     $$
>
>
> 2. On the other hand, consider the case when $K=1$​​, then $D_k=D. Theorem 2 seems to fail under this scenario. Am I missing anything?
>
>    **Response:** Our paper considers the scenario that there are more than one client. Therefore, Theorem 2 does not apply to the degenerated case where $K=1$. We have clarified this point in the revision.
>
> 3. Convergence analysis seems to apply only in the setting where $E=1$​​.
>
>    **Response:** We adopt a standard framework of convergence analysis in federated learning [4,5,6]. The key assumptions for convergence are 1. the gradient of the centralized objective equals the (expected) sum of local gradients, and 2. the diversity of local gradients is bounded. Since multiple epochs do not violate those assumptions, we believe the convergence result can be shown for multiple epochs following the techniques used in [4].
>
>    [4] Haddadpour and Mahdavi, 2019. On the convergence of local descent methods in federated learning.
>
>    [5] Li et al., 2020. Federated optimization in heterogeneous networks.
>
>    [6] Wang et al., 2020. Tackling the objective inconsistency problem in heterogeneous federated optimization.
>
> 4. As the authors mentioned in Remark 9, the authors are using the loss as the surrogate for parity. That means what is actually optimized is not the fairness metric mentioned earlier (e.g. DP, EO). As a result, the convergence bound does not directly transfer to fairness guarantee. Hence, it's unclear how well the proposed optimization does in terms of provable fairness guarantee.
>
>    **Response:** The motivation for using a surrogate is that some fairness metrics take discrete values, such as statistical disparity, thus are computationally intractable for optimization. As a result, using a surrogate function for the fairness metric is the most widely used approach to enable mitigate this issue. This approach is adopted in almost all works targeting at improving FL model fairness, including the methods used in our experiments and mentioned in the past work. In summary, it is intriguing to explore the influence of using a surrogate, while it is still an open challenge for this area.

---

> ### Author Response · Authors · 2024-06-11
> **Response 2**
>
> 5. Insufficient comparison with prior works. The authors should add comparison with a few prior works[1,2,3] that empirically evaluate on DP and EO in FL settings.
>
>    [1] S. Cui, W. Pan, J. Liang, C. Zhang, and F. Wang, “Addressing algorithmic disparity and performance inconsistency in federated learning,” Advances in Neural Information Processing Systems, vol. 34, 2021
>
>    [2] Hu, S., Wu, Z. S., & Smith, V. (2024, April). Fair federated learning via bounded group loss. In 2024 IEEE Conference on Secure and Trustworthy Machine Learning (SaTML) (pp. 140-160). IEEE.
>
>    [3] Chu, L., Wang, L., Dong, Y., Pei, J., Zhou, Z., & Zhang, Y. (2021). Fedfair: Training fair models in cross-silo federated learning. arXiv preprint arXiv:2109.05662.
>
>    **Response:** We are happy to compare with the works mentioned by the reviewer.
>
>    First, the method in [1] is proposed to ensure local fairness, while our FedGFT is intended to mitigate the bias of the global model. Therefore, it is not appropriate to compare our method with [1] directly. For example, although Figure 3 in [1] reports the disparity of DP and EO, but those values are the **average disparity for each local client**, which do not reflect the disparity of the global model. We also note that [2] is based on the bounded group loss, which is a privacy notion different from ours and can be viewed as a relaxation of demographic parity (DP) or equalized odds (EO). While [3] targets global fairness as ours, we regret that we do not find public code released by the authors of [3] (The code repo posted in this paper has been deleted).
>
>    While the code of [2] and [3] are not released and we are thus unable to reproduce their results in the rebuttal period, to address your concern, we compared our results to the results reported in [2] and [3]. In particular, Figure 3 in [2] reports the values of DP and EO on the COMPAS dataset. The lowest DP and EO level achieved by [2] is 0.02 while the accuracy drop compared to FedAVG is 0.01. In contrast, our FedGFT can achieve DP and EO level at 0.005, and the accuracy drop compared to FedAVG is within 0.01. Therefore, FedGFT achieves a better fairness-accuracy trade-off than [2] on the COMPAS dataset. Similarly, Table 1 in [3] reports the values of EO on both COMPAS and Adult datasets. The lowest EO level achieved by [2] is 0.03 while the accuracy drop compared to FedAVG is within 0.01. As a result, FedGFT also achieves a better fairness-accuracy trade-off than [3].
>
> 6. Zeng et al. present a similar result between local vs global fairness. Could the authors comment on that?
>
>    **Response:** We thank the reviewer for your helpful suggestion. We have discussed Zeng et al. (2021) in the past work section. Below, we further elaborate our discussion and comparison with this work. In short, the scope and contribution of Zeng et al. (2021) is different from ours.
>
>    Theoretically, Zeng et al. (2021) studied the problem is federated learning helpful for decentralized fairness learning. In particular, they show that in terms of fairness-utility trade-off, training locally fair models without communicating with each other and assembling them is worse than LFT (as defined in our paper), and LFT is worse than centralized fair learning. **In contrast**, we analyze the relationship between the fairness of local models and the global model in federated learning, which is untouched by Zeng et al. (2021).
>
>    As for the algorithm, Zeng et al. (2021) proposed FedFB that combines federated learning with FairBatch (a fair training method originally proposed in the centralized setting). FedFB solves a bi-level optimization problem to optimize fairness criterion, where the outer loop adaptively chooses training batches that fairly represent the training data, and the inner loop optimizes model parameters. **In contrast,** our proposed FedGFT is inspired by our developed theory (Theorem 8), namely the gradient of global fairness can be obtained from local fairness-related quantities. As a result, FedGFT can seamlessly integrate with any conventional federated learning algorithms such as FedAvg and FedProx, by only slightly adapting the local objective. Compared to FedFB, FedGFT offers a simpler optimization procedure without the need of bi-level optimization. Moreover, our methods apply to a broader class of fairness metrics than FedFB. For example, both conditional equal opportunity (Beutel et al., 2019) and balance for positive and negative class (Kleinberg et al., 2017) are proper group-based fairness metrics and therefore can be optimized using FedGFT, while they do not meet the assumptions made by FedFB.
>
>    Beutel A, Chen J, Doshi T, et al. Putting fairness principles into practice: Challenges, metrics, and improvements[C]. Proceedings of the 2019 AAAI/ACM Conference on AI, Ethics, and Society. 2019
>
>    Kleinberg J, Mullainathan S, Raghavan M. Inherent trade-offs in the fair determination of risk scores[J]. arXiv:1609.05807, 2017.

---

### Author Response · Authors · 2024-06-10
**To all reviewers**

We sincerely thank all reviewers for their invaluable feedback, which has significantly contributed to enhancing the quality of our work. We are pleased to learn that our theoretical contributions are perceived as significant and novel. We also appreciate the constructive suggestions regarding the improvement of our experiments.

We are committed to addressing every comment. Besides replying to each comment, we have also uploaded a revised manuscript. The major changes are highlighted in blue color, including

1. Elaborate Definition 1.
2. Add the discussion on FedGFT’s communication/computation cost, and extension to multi-class labels and involvement of partial clients.
3. Reorganized the entire experiment section.
   - Rerun the experiments to ensure that all methods in comparison reach convergence.
   - Clarify the influence of hyper-parameter for FedGFT.
   - Conduct additional experiments on different levels of data and client heterogeneity, including pure clients from a single group, uneven sample size per client, and large number of clients.
4. Add references related to group fairness in federated learning.
5. Clarifications on proof and experiments details.

Thank you again for your time and effort in reviewing our work.


Best,

Authors of Submission 2690

---

### Decision · Action_Editor_FyXL · 2024-07-23

**Recommendation:** Accept with minor revision

**Comment:**

This paper studies group fairness in federated learning, and explores the relationship between local group fairness (focusing on statistical parity and equal opportunity notions) and global group fairness. The main contribution of the paper is that it proposes a regularized objective for achieving global fairness, an algorithm for solving it with a single local epoch, and analyzing its convergence guarantees. The paper also provides results on analysis of differences between local and global model fairness notions.

The paper was reviewed by four experts on group fairness in federated learning. During the review process, key pieces of related work were identified by reviewers that had not been sufficiently discussed in the paper. After major revisions, all reviewers believe that the paper could be published given the empirical and theoretical evidence. However, the writing needs to still be revised to put the work in better perspective with respect to the related work. In particular, the work [1]-[4] needs to be discussed up front in the introduction, and the claims in the introduction, such as "We study the relationship between fairness of local and global models, for the first time revealing their underlying theoretical connection" need to be removed and replaced with statements that connect with these pieces of related work.

[1] Hamman, F., Dutta, S. Demystifying Local & Global Fairness Trade-offs in Federated Learning Using Partial Information Decomposition, 2023.

[2] Zeng, Y., Chen, H., & Lee, K. Improving fairness via federated learning, 2021.

[3] Hu, S., Wu, Z. S., & Smith, V. Fair federated learning via bounded group loss.

[4] Chu, L., Wang, L., Dong, Y., Pei, J., Zhou, Z., & Zhang, Y. Fedfair: Training fair models in cross-silo federated learning.

Additionally, Reviewer wLz4 has identified the following important issue with the training that needs to be addressed and/or acknowledged in a limitations section: "the new training parameters for all approaches seem inconsistent and irrational. For example, for the smallest dataset (COMPAS, with approx. 7K data points), the local updates and learning rate being used are quite large compared to the other datasets. I also note that in general, the proposed approach is evaluated on datasets that are very small and/or have been retired [5]."

[5] Retiring Adult: New Datasets for Fair Machine Learning. Frances Ding, Moritz Hardt, John Miller, Ludwig Schmidt

The paper is recommended to be accepted subject to the authors making these critical changes to their camera-ready revision and providing a point-by-point response to this letter. Congratulations!

**Audience:**

The paper is of broad interest to the TMLR audience.

**Claims And Evidence:**

The claims are largely supported by empirical evidence and theoretical reasoning; there are some remaining edits that are needed with respect to the novelty claims.

---

> ### Author Response · Authors · 2024-07-31
> **Request for extension of revision period**
>
> Dear Action Editor and Reviewers,
>
> We would like to express our sincere gratitude for the time and effort you have dedicated to reviewing our paper. We are pleased to learn that our work has been accepted with minor revisions.
>
> To ensure we address all comments thoroughly and submit the highest quality camera-ready version, we kindly request a two-week extension of the revision period, with a new submission deadline of September 6th, if possible.
>
> Thank you very much for your understanding and consideration.
>
> Best regards,
>
> Authors of Submission 2690

---

> ### Author Response · Authors · 2024-08-29
> **Camera-ready version**
>
> We sincerely thank the action editor and all reviewers for their invaluable feedback, which has significantly contributed to enhancing the quality of our work. We are dedicated to addressing every comment in the following response, and the camera-ready version has been uploaded.
>
> 1. "In particular, the work [1]-[4] needs to be discussed up front in the introduction"
>
>    **Response:** Thank you for the suggestion. We have now moved the discussion of related works to the introduction. This section has been reorganized to highlight the contributions of existing works and clarify how our work compares to them.
>
> 2. "the claims in the introduction, such as "We study the relationship between fairness of local and global models, for the first time revealing their underlying theoretical connection" need to be removed and replaced with statements that connect with these pieces of related work."
>
>    **Response:** Thank you for the insightful comment. We have revised this sentence to: "Under our proposed group-based fairness notion, we investigate the relationship between the fairness of
>    local and global models". Additionally, we have strengthened the connection to existing work by: (1) Adding that "Our work supports the results in [Chu et al., 2021] that local fairness implies global fairness for homogeneous clients. Our work also provides a fairness analysis based on data heterogeneity, complementing the information theory-driven analysis presented in [Hamman & Dutta, 2023]." (2) Highlighting the contributions of existing works such as [Zeng et al, 2021, Hu et al., 2024] in the introduction, along with a comparison to our work.
>
> 3. "The new training parameters for all approaches seem inconsistent and irrational. For example, for the smallest dataset (COMPAS, with approx. 7K data points), the local updates and learning rate being used are quite large compared to the other datasets. "
>
>    **Response:** We thank the reviewer for their insightful comment. The three datasets used in this paper differ in learning difficulty, which influences the choice of proper hyper-parameters. For example, an easier task typically has a smoother and flatter optimization objective, making a larger learning rate more suitable. Since the COMPAS dataset is the simplest among the three, a large learning rate is used compared to the other datasets. However, we have also conducted experiments on both Adult and COMPAS datasets using varied learning rates and epochs. The experiment results consistently show that FedGFT outperforms other methods in comparison. We have incorporated this point into the revision.
>
> 4. "I also note that in general, the proposed approach is evaluated on datasets that are very small and/or have been retired [5]."
>
>    **Response:** Thank you for bringing this to our attention. We use the Adult and COMPAS datasets because they are widely employed in existing fairness research, including the methods we compare against. As the reviewer noted, [5] proposed to retire the Adult dataset and use a suite of new datasets derived from US Census surveys for fair machine learning research. We have acknowledged this point in the limitation section.